# NAD$^+$ depletion is central to placental dysfunction in an inflammatory subclass of preeclampsia

Fahmida Jahan[1], Goutham Vasam[2] , Yusmaris Cariaco[2], Abolfazl Nik-Akhtar[1], Alex Green[1] , Keir J Menzies[1,2,3] , Shannon A Bainbridge[2,4]

Preeclampsia (PE) is a hypertensive disorder of pregnancy and a major cause of maternal/perinatal adverse health outcomes with no effective therapeutic strategies. Our group previously identified distinct subclasses of PE, one of which exhibits heightened placental inflammation (inflammation-driven PE). In non-pregnant populations, chronic inflammation is associated with decreased levels of cellular NAD$^+$, a vitamin B3 derivative involved in energy metabolism and mitochondrial function. Interestingly, specifically in placentas from women with inflammation-driven PE, we observed the increased activity of NAD$^+$-consuming enzymes, decreased NAD$^+$ content, decreased expression of mitochondrial proteins, and increased oxidative damage. HTR8 human trophoblasts likewise demonstrated increased NAD$^+$-dependent ADP-ribosyltransferase (ART) activity, coupled with decreased mitochondrial respiration rates and invasive function under inflammatory conditions. Such adverse effects were attenuated by boosting cellular NAD$^+$ levels with nicotinamide riboside (NR). Finally, in an LPS-induced rat model of inflammation-driven PE, NR administration (200 mg/kg/day) from gestational days 1–19 prevented maternal hypertension and fetal/placental growth restriction, improved placental mitochondrial function, and reduced inflammation and oxidative stress. This study demonstrates the critical role of NAD$^+$ in maintaining placental function and identifies NAD$^+$ boosting as a promising preventative strategy for PE.

## Introduction

Preeclampsia (PE) is a complex hypertensive disorder that occurs during pregnancy (at ≥20 wk of gestation) (1). The disease affects almost 10 million women each year, making it one of the major causes of mortality and morbidity in the mother and baby (1). Unfortunately, there is no cure for PE. Removal of the affected organ, the placenta, is the only "cure," often resulting in premature delivery and associated iatrogenic outcomes (2). Historically, PE has been studied as a unique pathological entity. However, there is considerable heterogeneity in the PE patient population based on the clinical profile, onset of disease, pregnancy outcome, fetal growth impacts, and placental histopathology. It is now acknowledged that PE likely develops because of several distinct underlying etiologies (3, 4, 5, 6, 7). Our group has previously characterized three different subclasses of PE pathophysiology using a multiscale profiling approach that incorporated gestational parent/fetal clinical features and detailed placental profiling (molecular and histological). Distinct subclasses of PE disease identified included the following: (1) gestational parent–driven PE (PE1)—likely the result of underlying gestational parent predisposing factors (involving minimal placental pathology); (2) hypoxia-driven PE (PE2)—resulting from placental ischemia/reperfusion injury (~60% of the PE patient population); and (3) inflammation-driven PE (PE3)—resulting from aberrant inflammation at the gestational parent–fetal interface (3, 4). Existence of such disease subclasses necessitates the requirement for studies focused on a better understanding of subclass-specific disease processes and/or subclass-specific interventions. To date, most research on PE has focused almost exclusively on understanding the hypoxia-driven pathophysiology of PE, whereas the etiological relevance of parental factors and/or inflammatory processes has been given less attention. The current study aims to address this gap in knowledge, specifically focusing on a better understanding of placental dysfunction associated with inflammation-driven PE, and subclass-specific therapeutic interventions for this unique patient population.

Placental mitochondrial dysfunction has long been recognized as a key component of PE pathophysiology (8, 9, 10, 11, 12). Altered placental mitochondrial respiration and/or total content is observed in tissues collected from pregnancies that span the clinical spectrum of PE, associated with altered oxidative phosphorylation (OXPHOS) protein expression, mitochondrial structural damage, defects in fusion/fission dynamics, and increased oxidative stress

---

[1]Department of Biochemistry, Microbiology and Immunology, Faculty of Medicine, University of Ottawa, Ottawa, Canada   [2]Interdisciplinary School of Health Sciences, Faculty of Health Sciences, University of Ottawa, Ottawa, Canada   [3]Ottawa Institute of Systems Biology, University of Ottawa, Ottawa, Canada   [4]Department of Cellular and Molecular Medicine, Faculty of Medicine, University of Ottawa, Ottawa, Canada

Correspondence: shannon.bainbridge@uottawa.ca; kmenzies@uottawa.ca

(8, 13, 14, 15). There is strong evidence to suggest that placental mitochondrial dysfunction may be a common feature across all PE subclasses (8); however, a lack of subclass-specific research endeavors has limited our current understanding of this pathology, specifically in the context of inflammation-driven PE. In non-pregnant populations, several inflammatory disease conditions have been tightly linked to mitochondrial dysfunction, in large part driven by profound overactivity of NAD$^+$-consuming enzymes, leading to depletion of intracellular NAD$^+$ (16, 17, 18, 19, 20, 21, 22, 23, 24, 25). In the current study, this body of work will be expanded to determine whether NAD$^+$ depletion may likewise initiate placental mitochondrial dysfunction in inflammation-mediated PE.

NAD$^+$, a vitamin B3 redox metabolite, is a critical coenzyme central to metabolic function throughout the body (26, 27, 28). NAD$^+$ maintains mitochondrial health by regulating energy metabolism and by inducing pathways, such as the mitochondrial unfolded protein response (26, 27, 28). Enzymes such as ADP-ribosyltransferases (ARTs), sirtuins (SIRT), cluster of differentiation 38/157 (CD38/CD157), and sterile alpha and Toll/interleukin-1 receptor motif–containing protein 1 (SARM1) require NAD$^+$ as a co-substrate to carry out their enzymatic functions (26, 27, 28, 29). Among these, the most well-described NAD$^+$-consuming enzyme family is the ARTs, which includes the diphtheria toxin–like family (ARTDs) that perform most of the ADP-ribosylation in mammals and are often referred to using their old family nomenclature of PARPs (30). A primary function of PARPs is the post-translational modification of proteins, via the covalent addition of ADP-ribose polymer(s) to various proteins—a process known as mono-ADP-ribosylation (MARylation) when a single unit of ADP-ribose is added or polyADP-ribosylation (PARylation) when multiple ADP-ribose units are added (30). This NAD$^+$-dependent process helps to regulate numerous biological processes such as chromatin organization, mRNA stability, transcriptional control, DNA methylation, glycolysis, inflammation, immune response, and DNA damage repair (17, 30, 31, 32, 33, 34, 35, 36). Under healthy homeostatic conditions, the cellular NAD$^+$ content is tightly regulated, achieving an optimal balance between NAD$^+$ biosynthesis/salvage and NAD$^+$ consumption pathways. However, under pro-inflammatory conditions, some of the major NAD$^+$ consumers, such as PARPs, become highly activated, tilting the scales of NAD$^+$ homeostasis toward excessive consumption, quickly depleting intracellular NAD$^+$ stores, and leading to mitochondrial dysfunction and impaired energy metabolism (16, 17, 18, 20, 21, 22). Interestingly, in a mouse model of hypoxia-driven PE, replenishment of whole-body NAD$^+$ stores via nicotinamide (NAM) treatment has been shown to rescue several PE-like features (i.e., hypertension, fetal growth restriction) (37, 38). However, the role of placenta-specific NAD$^+$ depletion in the pathogenesis of PE and whether supplementation with an NAD$^+$ precursor will be an attractive preventative strategy for other PE subclasses are unknown.

In the current study, the potential role(s) of inflammation-mediated NAD$^+$ depletion and subsequent mitochondrial dysfunction in the establishment of placental disease in PE were explored using complementary studies on human placental tissues, in a human trophoblast cell culture model and in a rat model of inflammation-driven PE. Furthermore, the therapeutic potential of NAD$^+$ boosting, via nicotinamide riboside (NR) (39) treatment, on placental health, fetal growth, and clinical manifestations of inflammation-mediated PE was evaluated. The overarching hypothesis tested was that placental NAD$^+$ depletion, resulting from inflammation-mediated hyperactivation of NAD$^+$-consuming enzymes, leads to impaired placental mitochondrial function and subsequent placental disease in the inflammatory subclass of PE. Furthermore, it is proposed that replenishing placental NAD$^+$ stores could be an attractive preventative strategy for this distinct subclass of PE patients.

# Results

## Inflammation-driven PE demonstrates evidence of NAD$^+$ depletion and mitochondrial dysfunction

Previous gene set enrichment carried out by our group using a genome-wide microarray dataset from human placental samples annotated according to the PE subclass (Table 1) identified an overexpression of several pro-inflammatory signaling pathways uniquely within one PE subclass—labeled inflammation-mediated PE (PE3) (3, 4). We further confirmed the overexpression of pro-inflammatory markers such as *TNF* super family members, *IFNG*, *IL1A, IL1B, IL6R,* and *IL17A,* as well as related mediators, exclusively in PE3 placentas (Fig 1A).

PARP hyperactivation has been widely described in non-pregnant inflammatory conditions, leading to excess protein ADP-ribosylation and NAD$^+$ depletion (16, 17, 18, 20, 21, 22). As such, the degree of protein ADP-ribosylation was assessed in human placental tissues collected from each of the three PE subclasses, along with term, preterm, and chronic hypertension controls. Global protein ADP-ribosylation was also found to be uniquely higher in the placentas from PE3, compared with other PE subclasses and all control groups (Fig 1B and C). In parallel, LC/MS measurements of NAD$^+$ and NADH in these same tissues demonstrated significant depletion of NAD$^+$ and total NAD(H) in PE3 placentas, compared with control preterm and term placentas (Fig 1D and E). No significant decrease was observed for the NAD$^+$ and total NAD(H) content in the other PE subclasses (PE1 and PE2) or in the chronic hypertension groups (Fig 1D and E). NADH levels alone, the NAD$^+$/NADH ratio, and NAM, an end product of NAD$^+$ consumption, on the contrary, did not change in any of the groups (Figs 1F and G and S1A).

Placental mitochondrial dysfunction and oxidative stress have been described as key contributors to placental dysfunction in PE when considered as a single clinical disease (8, 9, 10, 11, 12). Thus, we aimed to determine whether a depleted NAD$^+$ pool in PE3 placentas may be associated with placental mitochondrial dysfunction. We found no indications of an altered placental mitochondrial content between any of the three PE subclasses and all control groups when using citrate synthase expression and mitochondrial complex IV (COX-IV) activity assays as surrogates of the mitochondrial content (40, 41) (Fig S1B and C). However, using a proteomic approach on a subset of placental samples, numerous mitochondrial proteins were found to be down-regulated in PE3 placentas compared with controls, including: subunits of OXPHOS protein complexes (NADH:Ubiquinone Oxidoreductase Subunit V3 [NDUFV3], Ubiquinol-Cytochrome C Reductase Core Protein 1 [UQCRC1], and

**Table 1.   Study population demographics.**

|  | Control (preterm and term) (n = 43) | Chronic hypertension (n = 23) | PE1 (n = 16) | PE2 (n = 42) | PE3 (n = 17) |
|---|---|---|---|---|---|
| Maternal age[a] | 31.4 ± 4.7 | 36.1 ± 4.6[b] | 32.2 ± 5.4 | 33.1 ± 6.3 | 31.4 ± 4.2 |
| Gestational age at delivery (weeks)[a] | 34.1 ± 5.3 | 33.2 ± 4 | 35 ± 3.1 | 31 ± 3.1[b] | 33.3 ± 2.80 |
| Fetal sex (F)[c] | 21/43 | 14/23 | 7/16 | 23/42 | 7/16 |
| Birthweight (g)[a] | 2,481.3 ± 1,115.09 | 1,649.8 ± 867.7[d] | 2,292.2 ± 813.1 | 1,290.6 ± 611.1[e] | 1707.7 ± 602.7[b] |
| Birthweight %ile (<10[th] %ile)[c] | 2/43 | 15/23 | 5/16 | 30/42 | 10/17 |
| Placental weight (g)[a] | 541.09 ± 180.4 | 344.1 ± 164.8[e] | 454.7 ± 133.5 | 312.3 ± 128.2[e] | 372.7 ± 147.1[d] |
| Placenta hypoplasia (<10[th] %ile)[c] | 0/43 | 9/23 | 1/16 | 15/42 | 4/16 |
| Maximum maternal systolic blood pressure[a] | 118.4 ± 15.4 | 162.8 ± 19.6[e] | 158.3 ± 15[e] | 162 ± 16.9[e] | 150.7 ± 15.6[e] |
| Maximum maternal diastolic blood pressure[a] | 71.9 ± 11.6 | 101 ± 11.6[e] | 100.7 ± 7.8[e] | 103.7 ± 12.8[e] | 95.8 ± 11.3[e] |
| Mode of delivery (C-section)[c] | 21/43 | 20/23 | 9/16 | 40/42 | 16/17 |

[a]Mean ± SD.
[b]$P < 0.01$.
[c]Number of pregnancies out of total number per group.
[d]$P < 0.001$.
[e]$P < 0.0001$.

UQCR Rieske Iron-Sulfur Polypeptide 1[UQCRFS1]), and proteins that affect mitochondrial dynamics (optic atrophy type 1 [OPA1]), mitochondrial import (TOMM7), and mitochondrial translation (mitochondrial ribosomal proteins, MRPS36, MRPL44, MRPS36, MRPL22, MRPL1), among others (Fig 1H). Placental samples from cases of hypoxia-driven PE (PE2) likewise demonstrated a decrease in some mitochondrial proteins compared with controls, such as OXPHOS protein–ATP synthase membrane subunit f (ATP5J2), mitochondrial translation elongation factor (TSFM), MRPL44, alongside other markers (Fig S1D). However, placentas from parental-driven PE (PE1) did not show any changes in mitochondrial protein profiles compared with healthy controls.

Mitochondrial impairment is thought to be linked to oxidative stress and placental dysfunction in PE (8, 9, 10, 11, 12); however, this relationship has not been explicitly examined in a subclass-specific manner. For this purpose, we performed 8-oxo-2′-deoxyguanosine (8-oxo-dG) staining—detecting oxidized guanine residues in DNA (42)—of placental sections collected from all three PE subclasses. Interestingly, both hypoxia-driven PE (PE2) and inflammation-driven PE (PE3) exhibited increased levels of oxidized nucleotides, when compared to parental-driven PE (PE1) and controls (Fig 1I and J).

### Boosting NAD⁺ prevents human trophoblast cell dysfunction under inflammatory conditions

To examine whether inflammation-mediated NAD⁺ depletion alters human trophoblast health and function, a TNF-α–induced inflammatory insult was applied to the HTR8/SVneo trophoblast cell culture model—representative of the invasive extravillous cytotrophoblast population, long considered a key trophoblast population contributing to the establishment of PE pathophysiology (43, 44, 45, 46). After 24 h of treatment with 10 ng/ml of TNF-α, we observed a significant increase in protein ADP-ribosylation (Fig 2A and B), coupled with a decrease in total cellular NAD⁺ levels and the NAD⁺/NADH ratio (Fig 2C). Interestingly, when an intracellular NAD⁺ content was boosted via NR treatment (150 µM–1 mM; Fig S2), despite the continued presence of an inflammatory insult (TNF-α treatment), excessive protein ADP-ribosylation was attenuated (Fig 2A and B). This would indicate that higher intracellular NAD⁺ concentrations are either inhibiting inflammation-mediated protein ADP-ribosylation or promoting faster flux of protein (de)ADP-ribosylation. To determine whether inflammation-mediated NAD⁺ depletion had any effect on mitochondrial function in this model, an XFe96 Seahorse assay was conducted to measure mitochondrial oxygen consumption rates (OCRs) (46). TNF-α treatment caused a decrease in basal and ATP-linked respiration, whereas NAD⁺ boosting with NR co-treatment was able to rescue these respiration rates (Fig 2D and E). No difference in maximal respiratory capacity was observed with either TNF-α or NR co-treatment (Fig 2F). Overall, these data suggest that TNF-α treatment induces trophoblast mitochondrial respiratory dysfunction, a phenotype that can be rescued by NR treatment. This finding of TNF-α–mediated mitochondrial dysfunction is consistent with previous findings, showing that TNF-α treatment during pregnancy reduces placental mitochondrial efficiency (47). Simultaneously, the glycolytic capacity of these cells was assessed via measurements of extracellular acidification rates. However, there was no significant decrease in glycolytic capacity in any treatment groups (Fig 2G). An assessment of the expression of key OXPHOS subunit proteins (succinate dehydrogenase B [SDHB], Ubiquinol-Cytochrome C Reductase Core Protein 2 [UQCRC2], Mitochondrially Encoded Cytochrome C Oxidase I [MTCO1], ATP synthase subunit alpha of complex V [vATP5A]), revealed no change in the protein content with NR co-treatment, suggesting the improved mitochondrial respiration rates observed are likely related to mitochondrial functional capacity

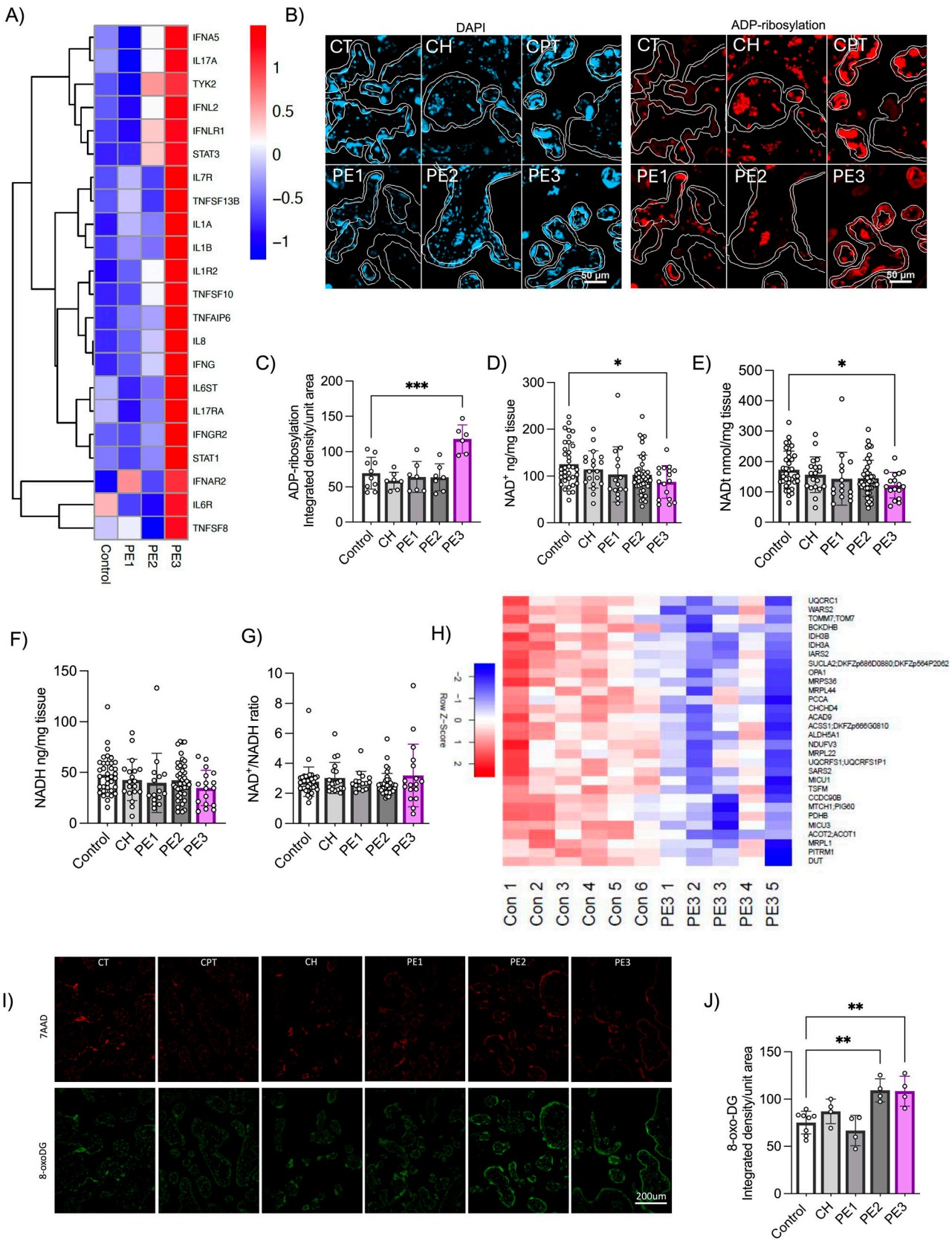

improvements rather than mitochondrial content improvements (Fig 2H–L). Finally, an assessment of the impact of inflammation-mediated NAD⁺ depletion on trophoblast function was carried out using a Matrigel-coated Boyden chamber invasion assay. The primary function of the extravillous cytotrophoblast cell lineage is invasion of the decidua and remodeling of the uterine spiral arterioles, ensuring adequate uteroplacental blood supply to support fetal growth (48). Previous studies have demonstrated an inhibitory effect of TNF-α treatment on the invasive capacity of these cells using either the HTR8 cell culture model or the first-trimester ex vivo placental explants (49, 50, 51). We were able to replicate these findings, demonstrating a 50% reduction in trophoblast invasion with TNF-α treatment. Importantly, intracellular NAD⁺ boosting with NR co-treatment rescued this deficit, returning the invasive capacity of these inflammatory insult–exposed cells back to those observed in untreated controls (Fig 2M and N). Collectively, these cell culture findings provide preliminary evidence, which suggests that NAD⁺ replenishment via NR treatment may improve human trophoblast health and function under pro-inflammatory conditions.

## Boosting NAD⁺ levels during pregnancy improves pregnancy outcomes in an inflammation-driven PE model

As NAD⁺ depletion was observed in placentas from cases of inflammation-driven PE, and NAD⁺ supplementation in vitro improved human trophoblast health and function under pro-inflammatory conditions, we next sought to determine whether NR could be used to prevent the development of PE-like clinical features in a rat model of inflammation-driven PE. Our group has previously characterized and compared several rat models of inflammation-mediated PE, determining whether the LPS-induced rat model was able to most closely recapitulate features observed in cases of human inflammation-driven PE (52). In this model, LPS (20–70 μg/kg/day) was injected intraperitoneally from gestational days (GD) 13–18. In our therapeutic intervention group, NR was administered by oral gavage from GD 1–19 (200 mg/kg/day) (Fig 3A). As previously described (52), maternal BP was heightened at the end of pregnancy with LPS treatment compared with saline-treated controls (95.77 ± SD versus 80.02 ± SD at GD 19, respectively); however, this phenotype was rescued with NR intervention (74.13 ± SD at GD 19) (Fig 3B).

In the LPS-treated pregnancies, there was a significant decline in fetal survival to term (Fig 3C), with the remaining viable fetuses demonstrating evidence of fetal growth restriction (Fig 3D). Importantly, these adverse fetal outcomes were attenuated with NR intervention (Fig 3C and D). Placental weights were decreased with LPS treatment (Fig 3E), with morphometric analysis demonstrating a

decrease in total placenta area (Fig 3F and G), largely attributed to the decreased size of the labyrinth compartment of the placenta (Fig 3H)—the site of maternal–fetal exchange. NR intervention during pregnancy was therefore able to rescue this adverse LPS-induced placental phenotype. On the contrary, the area of the junctional zone (the hormone production area of the placenta) and the maternal decidua did not change with any treatment group (Fig 3I and J).

## Boosting NAD⁺ levels during pregnancy improves placental health and function in an inflammation-driven PE model

To determine whether LPS treatment can induce placental inflammation, we first examined placental TNF-α levels by ELISA. TNF-α is considered to be a key contributor to the pathogenesis of hypertension in pregnancy (52, 53, 54), and according to our microarray data, the TNF-α pathway is up-regulated in the inflammatory PE3 placentas (Fig 1A). Similarly, we show that LPS treatment induces rat placental TNF-α expression, which can be normalized with NR intervention (Fig 4A). RNA-seq analysis performed on a subset of placentas further confirms that NR treatment in the presence of LPS modulates the placental transcriptome (Fig S3A–D) and down-regulates the defense response and response to a chemokine (Fig S3D) suggesting a reduction in inflammation.

To test whether placental ADP-ribosylation is hyperactivated in this rodent model of inflammatory PE, and whether it is associated with a decline in the placental NAD⁺ content, we measured the total protein ADP-ribosylation and NAD⁺ content in the collected placentas. We found that LPS treatment led to increased placental protein ADP-ribosylation, whereas NR intervention normalized this change (Fig 4B and C). The levels of ADP-ribose (ADPR), produced from the hydrolysis of ADP-ribosylation units by ADP-ribosyl-acceptor hydrolases (55, 56), were higher in placentas from the LPS-induced PE model and normalized with NR treatment, indicating that NR intervention likely did not increase ADP-ribosylation hydrolysis to result in lower ADP-ribosylation levels (Fig 4D). We further found a decrease in NAD⁺, NADt (NAD⁺ and NADH), NADH, and the NAD⁺/NADH ratio in placentas from LPS-treated pregnant rats, compared with saline controls, with NR intervention preventing such declines in NAD⁺, NADt (NAD⁺ and NADH), and NADH levels (Fig 4E–G). However, the NAD⁺/NADH ratio was not significantly rescued with NR supplementation (Fig 4H). Given that NR alone did not alter NAD(H) levels in the rat placentas, we measured the levels of NAM, which is a by-product of NAD⁺-consuming enzymatic reactions, which can be converted back to NAD⁺ by the salvage pathway (31). As expected, we found high levels of NAM in both NR alone and NR with LPS-treated placentas (Fig S4A).

**Figure 1. Identification of altered NAD⁺/ADP-ribosylation signaling, mitochondrial proteomes, and oxidative damage in human placentas with inflammation-driven PE (PE3).**
**(A, B, C)** Gene expression of inflammatory markers in human placenta biopsies from all three PE subclasses (n = 4 placentas/group). (B, C) Representative immunofluorescence images and quantification of protein ADP-ribosylation in trophoblasts of human placental tissue sections from all three PE subclasses and various control groups (n = 7–8 placentas/group). ADP-ribosylation was measured specifically within the trophoblasts highlighted using the white lines. **(D, E, F, G)** LC/MS quantification of human placental NAD(H) levels (n = 15–42 placentas/group). **(H)** Heatmap of down-regulated mitochondrial proteins in inflammatory PE3 (term) placentas compared with control term placentas (n = 5–6 per group). Shown are only proteins that met the false discovery rate threshold of 0.05. **(I, J)** Placental oxidative stress was determined by immunofluorescence staining of human placental tissue sections with anti-8-oxo-dG antibody and quantification by the integrated intensity/area of trophoblasts (n = 4 per group). One-way ANOVA with Holm–Šidák's multiple comparisons test, *P < 0.05, **P < 0.01, ***P < 0.001. The error bar indicates the SD. Control = control term and preterm, CH = chronic hypertension control, PE1 = gestational parent–driven PE subclass, PE2 = hypoxia-driven PE subclass, PE3 = inflammation-driven PE subclass.

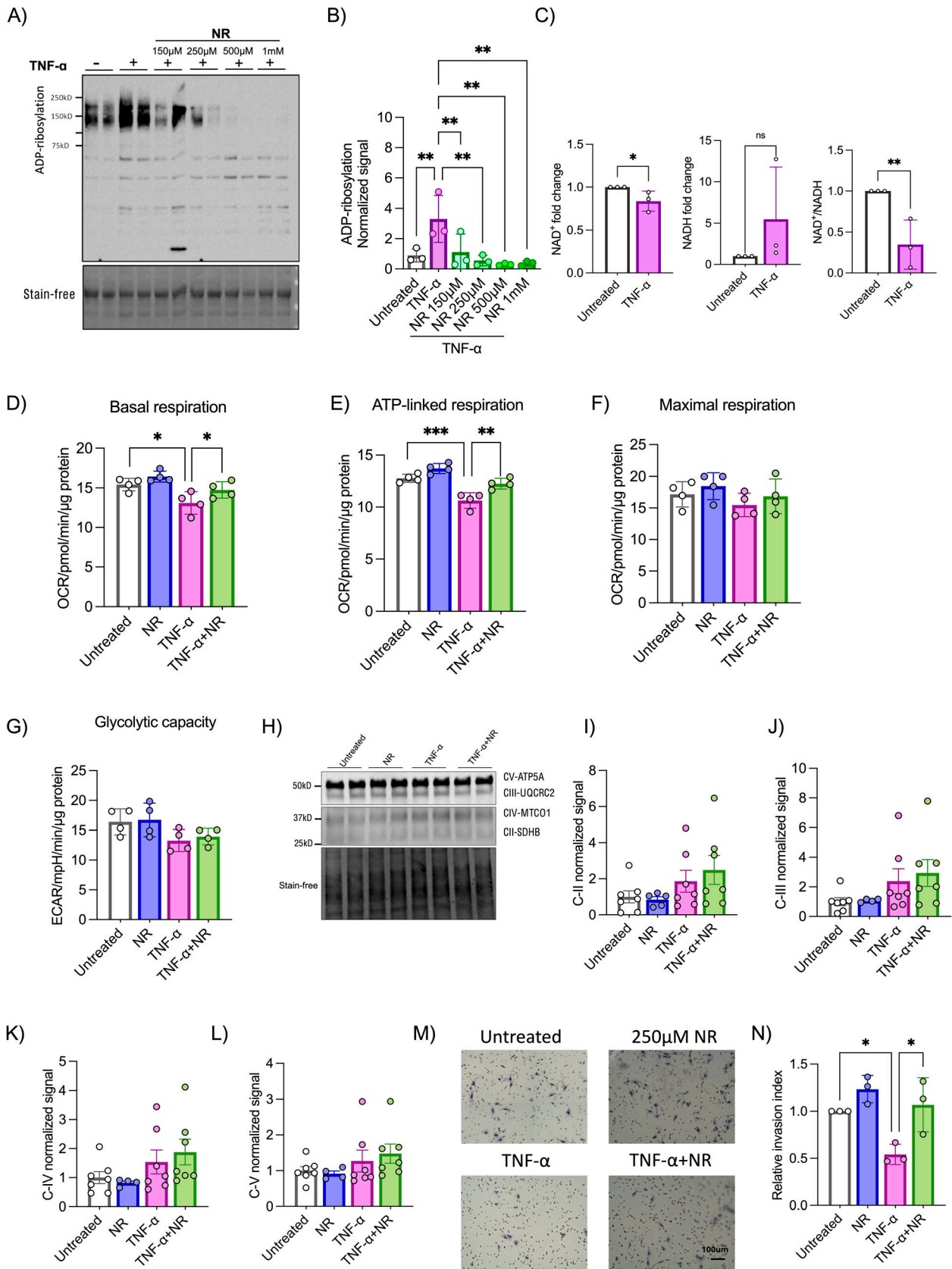

To determine whether LPS treatment leads to placental mi-tochondrial dysfunction and whether NR exerts its beneficial effect in part by improving mitochondrial function, we performed oxygraph respirometry on freshly isolated mitochondria from these placentas. LPS treatment decreased complex I– and complex II–driven state-III respiration rates, whereas NR inter-vention normalized respiration rates (Fig 4I and J). Complex IV–mediated respiration was unaltered by LPS or NR treatments (Fig S4B). Similar to measurements in human HTR8 cells, these functional changes were not a result of the altered expression of NADH:Ubiquinone Oxidoreductase Subunit B8 (NDUFB8), UQCRC2, or vATP5A OXPHOS proteins (Fig S4C–F). Given that the placental mitochondrial fusion is impaired in PE (8) and that the expression of the fusion protein OPA1 was reduced in human inflammatory PE placentas (Fig 1H), we examined the expression of OPA1 in all conditions. Although OPA1 protein expression is significantly reduced in placentas from LPS-treated rats, NR intervention did not attenuate this effect (Fig S4G and H). A compromised mitochondrial quality control pathway has been shown to contribute to placental dysfunction in canonical/ hypoxia-driven PE2 (53).

Thus, we determined the expression levels of two mito-chondrial quality control proteases—YME1 like 1 ATPase (YME1L1) and Caseinolytic Mitochondrial Matrix Peptidase Proteolytic Subunit (CLPP). Both YME1L1 and CLPP expression levels were unaltered in each of the different treatment conditions (Fig S4I–K).

As chronic inflammation and mitochondrial dysfunction can result in oxidative stress (54, 55, 56, 57) and our results show that human inflammatory PE placentas exhibit more oxi-dative DNA damage (Fig 1I and J), we next investigated whether LPS treatment leads to placental oxidative DNA damage in this model. We performed Western blotting to determine the levels of the phosphorylated form of histone H2A.X (pH2A.X), a marker of a DNA double-stranded break (58). Placentas from LPS-treated rats exhibit higher levels of the pH2A.X protein, whereas NR intervention significantly attenuated this effect (Fig 4K and L). We further confirmed this by immunohis-tochemistry, showing that LPS-treated placentas have more pH2A.X-positive cells in the labyrinth exchange region and the maternal decidua, whereas NR intervention normalized this effect to control levels (Fig 4M and N). This is consistent with studies that have shown that NAD+ replenishment can lead to decreased pH2A.X levels, indicative of a reduction in DNA damage (59).

## Discussion

We demonstrate for the first time that NAD+ levels are depleted in both human and rat placentas affected by inflammation. Our findings suggest that this reduction in NAD+ may be driven by the hyperactivity of NAD+-consuming PARP enzymes. In vitro, we have shown that boosting NAD+ levels with NR can improve trophoblast health and function under inflammatory conditions. Using the LPS-induced inflammatory PE rat model, we demonstrated that oral pre-treatment with NR prevents the development of PE-like clinical features, evidenced by lowered maternal blood pressure, de-creased placental inflammation, increased placental and fetal weights, and increased fetal survival.

We have observed that increased protein ADP-ribosylation is unique to human inflammation–driven PE (PE3) placentas, findings that were replicated in placentas from LPS-treated pregnant rats. Currently, we do not know which PARPs are important in the context of inflammatory PE. Among the 17 members of the PARP family, PARP1/2/5a/5b are known to have PARylating activity, whereas the rest (PARP3/4/6/7/8/10/11/12/14/15/16) are associated with mono-ADP-ribosylation (MARylation). In contrast, PARP9/13 have no enzymatic activity (34). Among these, PARP1—a major NAD+ consumer of the cell—plays a critical role in inflammation (60, 61, 62, 63, 64, 65, 66). PARP1 is required for the nuclear factor kappa-light-chain-enhancer of activated B (NF-kB) transcription and its acti-vation (67). It also increases mRNA stability of pro-inflammatory genes and enhances the release of pro-inflammatory molecules such as high-mobility group box protein 1 (HMGB1) (68, 69). MAR-ylating and non-enzymatic PARPs, such as PARP9/11/12/13/14, have broad-spectrum antiviral properties, and their gene expression is known to increase under conditions of inflammation and reduce the tissue NAD+ content (17, 70). Apart from having an inflammatory signature, we also observed oxidative DNA damage in human and rat inflammation-driven PE placentas. The PARylation activity of both PARP1 and PARP2 recruits DNA damage proteins to sites of DNA damage (34). Thus, it is likely that various PARylating and MAR-ylating enzymes are responsible for the decline in placental NAD+ levels observed in this work. Because of the substantial contri-bution of PARP1 in inflammation, its inhibition has been shown to reduce inflammation and restore cellular NAD+ levels in several studies (63, 64, 65, 66). However, PARP1 inhibition or knockout in pregnant rodent models has caused profound fetal defects, sug-gesting its essential role during fetal development (71, 72, 73, 74). Several studies suggest that PARP1 is required for cell survival under mild oxidative stress, as it participates in DNA damage repair

**Figure 2. Potential of NAD+ boosting to improve HTR8/SVneo human trophoblast health and function under an inflammatory in vitro condition.**
**(A, B)** Trophoblast protein ADP-ribosylation was determined by Western blot with anti-ADP-ribosylation antibody on cells treated with 10 ng/ml TNF-α and/or 150 μM–1 mM NR for 24 h (n = 3). One-way ANOVA with Holm–Šídák's multiple comparisons test. **(C)** Total cellular NAD+(H) and NAD+/NADH ratio quantification was performed on cells treated with or without 10 ng/ml TNF-α for 24 h (n = 3). One-tailed unpaired *t* test. **(D, E, F, G)** XFe96 Seahorse Mito stress test was performed on cells treated with 10 ng/ml TNF-α and/or 250 μM NR for 24 h. Basal, ATP-linked, and maximal mitochondrial respiration rates and extracellular acidification rates were measured (n = 4). **(H, I, J, K, L)** Expression of oxidative phosphorylation (OXPHOS) proteins was determined by Western blot with OXPHOS antibody cocktail on cells treated with 10 ng/ml TNF-α and/or 250 μM NR for 24 h (n = 4–7). **(M, N)** Trophoblast invasion capacity was determined by performing a Matrigel invasion assay on cells treated with 10 ng/ml TNF-α and/or 250 μM NR for 72 h (n = 3). Two-way ANOVA with Holm–Šídák's multiple comparisons test, *P < 0.05, **P < 0.01, ***P < 0.001. The error bar indicates the SD. TNF-α = tumor necrosis factor, NR = nicotinamide riboside.
Source data are available for this figure.

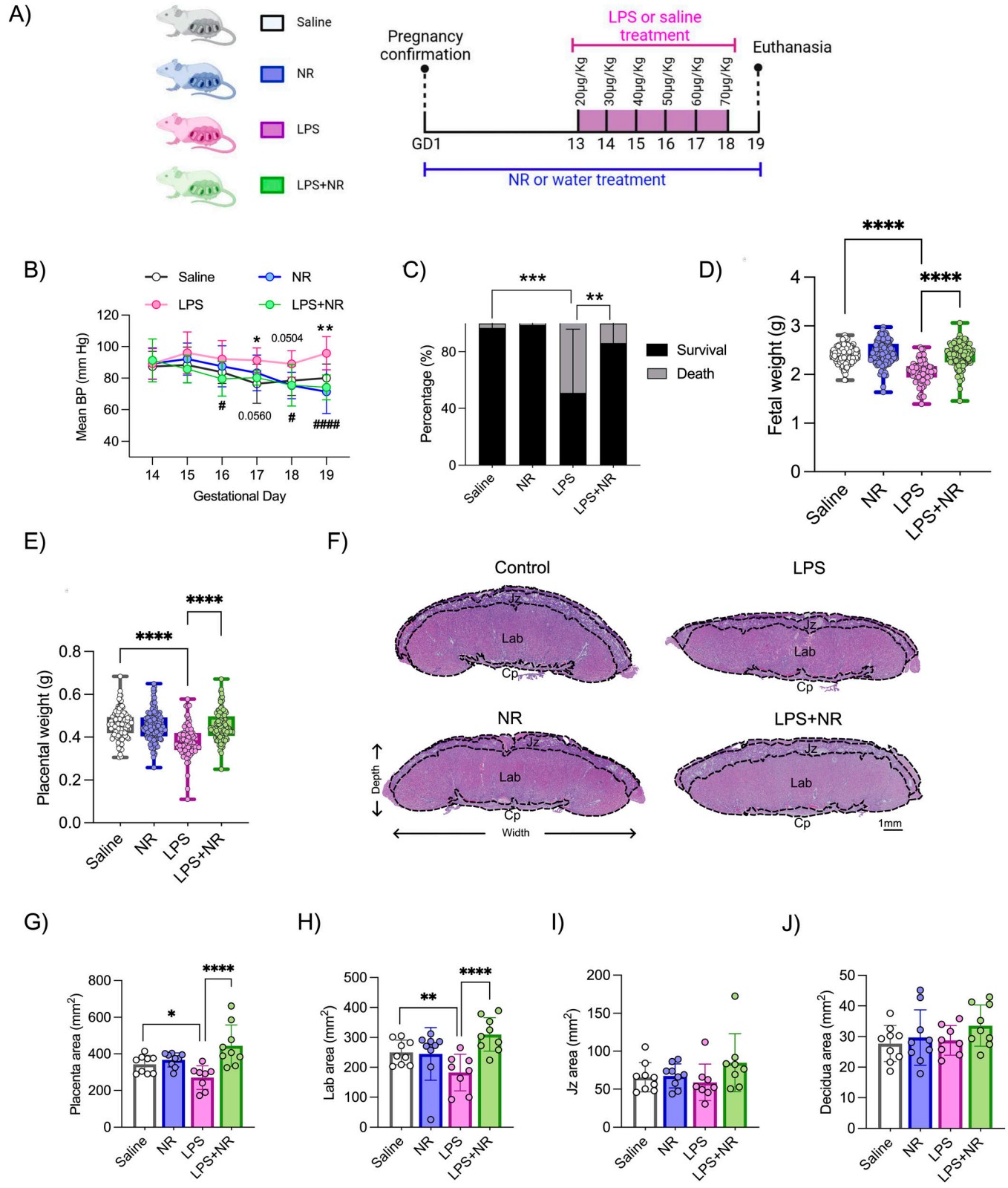

**Figure 3. Effect of NAD⁺ boosting in preventing inflammation-driven PE in a rat model.**
**(A)** Study design: pregnant rats were given either NR or drinking water from GD 1–19, and either LPS (20–70µg/kg/day) or saline was injected from GD 13–18; euthanasia was performed on GD 19 (created with BioRender.com on 12 June 2023). **(B)** Blood pressure was measured from GD 14–19 by the tail–cuff method (n = 6–8 pregnant rats/treatment). * indicates *P*-values between saline and LPS, and # indicates *P*-values between LPS and LPS+NR. **(C)** Fetal survival and death percentage. Two-way ANOVA with

(75, 76, 77). As placental oxidative stress is observed in PE, it is likely that activation of PARP1 is required for the adaptation to this stress. As such, boosting NAD$^+$ levels presents an effective and safe treatment option for supporting ADP-ribosylation signaling in inflammation-driven PE. Follow-up studies will be important to better understand and tease apart the unique roles of various PARPs in the context of PE.

It is now well established that depletion of the NAD$^+$ content is a hallmark of defective energy metabolism (78). This is now recapitulated in the placenta, where our study suggests that chronic inflammation in the placenta leads to a decline in NAD$^+$ availability and mitochondrial dysfunction. Currently, we do not know the exact mechanism of how NAD$^+$ boosting improves placental mitochondrial function. Under pro-inflammatory conditions, we observed an improvement in ETC function with NR supplementation in trophoblast cells or placental tissue but not in the mitochondrial content. We speculate that because energy metabolism is dependent on NAD$^+$ and its reduced form NADH, a decline in the total NAD$^+$ content will lead to depressed energy metabolism, and NAD$^+$ supplementation will prevent that. Moreover, activation of the SOD2/SIRT3 pathway could be a potential mechanism of reducing mitochondrial ROS and improving placental function with NR supplementation, which needs further exploration.

NAD$^+$ boosting has been proven effective in various pro-inflammatory disease models (16, 17, 18, 29, 59, 79, 80). Moreover, NR has been shown to be orally bioavailable and safe for human consumption (25, 81, 82, 83). Several non-pregnant human studies, including those involving healthy, obese, or overweight individuals, and patients with ataxia, Alzheimer's, or Parkinson's disease, have demonstrated that NR is effective in elevating NAD$^+$ levels and can improve systemic metabolic function (25, 84, 85, 86, 87, 88, 89). NR was also shown to reduce circulating inflammatory markers in humans (83), suggesting that it could be particularly effective for inflammation-driven PE patients. NAM has been found effective in hypoxia-driven PE mouse models (37, 38) preventing hypertension and fetal growth restriction by reducing glomerular endotheliosis and improving the metabolic profile of the fetal brain (37, 38). Thus, there is a strong argument that boosting NAD$^+$ levels may combat all subclasses of human PE (8), thus making it a very promising therapeutic intervention.

In conclusion, we showed for the first time that NAD$^+$ plays a critical role in maintaining healthy placental function. We have mechanistically shown that depleted placental NAD$^+$ levels are a signature of human inflammatory PE and demonstrated that increasing NAD$^+$ levels enhances placental function and thus prevents the development of the PE3 subclass. Future pre-clinical and clinical trials could test the effectiveness of NAD$^+$ boosting strategies, such as supplementing with vitamin B3 derivatives, during pregnancy or once PE has been diagnosed to establish its treatment potential.

# Materials and Methods

### Human sample processing

All human tissue experiments were carried out according to the University of Ottawa and Mount Sinai Hospital Research Ethics Board guidelines and approval (University of Ottawa REB protocol# H08-17-08, Mount Sinai Hospital REB protocol# 20-0037-E). For the measurement of the placental NAD$^+$ and mitochondrial content, a total of 128 frozen placental samples and matched histological sections were purchased from the Research Centre for Women's and Infants' Health BioBank (RCWIH), Mount Sinai Hospital, Toronto, Canada. Samples included term and preterm placentas from healthy controls (n = 43), individuals with chronic hypertension (n = 23), and gestational parent–driven PE subclass (PE1; n = 16), hypoxia-driven PE subclass (PE2; n = 42), and inflammation-driven PE subclass (PE3; n = 17) patients. PE was defined as the onset of hypertension (systolic pressure > 140 mm Hg and/or diastolic pressure > 90 mm Hg) after 20 wk of gestation coupled with evidence of maternal end-organ dysfunction (2). The classification of the molecular subclass was done by microarray (confirmed by qRT-PCR analysis) and histopathology analysis, as previously described (3, 90). For comparison purposes, control healthy term and preterm placentas were combined as a singular healthy "control" group, because PE placentas included both term and preterm placentas.

### Microarray analysis of gene expression
A microarray dataset previously generated by our group and available at the Gene Expression Omnibus (GEO) database (GSE75010) was used to assess the PE subclass-specific expression of inflammatory markers and NAD$^+$-consuming enzymes (3). This microarray dataset was generated from frozen human placental tissues that were purchased from RCWIH, Mount Sinai Hospital, Toronto, Canada. Briefly, RNA isolation was performed using TRIzol and RNeasy spin columns. Human Gene 1.0 ST Array chips (Affymetrix) were used to perform microarray at the Princess Margaret Genomics Centre (Toronto, Canada). The heatmap of the expression of key genes of interest was generated in RStudio.

### Measurement of ADP-ribosylation
Paraffin-embedded placental tissue sections (5 $\mu$m), mounted on glass slides, were purchased from RCWIH, Mount Sinai Hospital, Toronto, Canada. These specimens were collected from the same cases as the frozen tissue biopsies described above (Table 1) and were used to assess protein ADP-ribosylation and oxidative stress (see the Measurement of oxidative stress section below) within the placenta for each of the three PE subclasses and controls. Sections were dewaxed in xylene, and hydrated in a sequence of 100%, 95%, 80%, 70%, and 50% ethanol, and water. Sections were subjected to antigen retrieval using citrate buffer (0.1 M, pH 6.0) in a microwave

---

Holm–Šídák's multiple comparisons test. **(D, E)** Fetal and placental weights were measured (n = 10–12 litters/treatment). One-way ANOVA with Holm–Šídák's multiple comparisons test. **(F, G, H, I, J)** Total placenta area along with individual regions such as labyrinth (Lab), junctional zone (Jz), and decidua was measured (n = 7–9 litters/treatment, 1 placenta/litter). Two-way ANOVA with Holm–Šídák's multiple comparisons test, */#$P$ < 0.05, **$P$ < 0.01, ***$P$ < 0.001, ****/####$P$ < 0.0001. The error bar indicates the SD. LPS = lipopolysaccharide, NR = nicotinamide riboside.

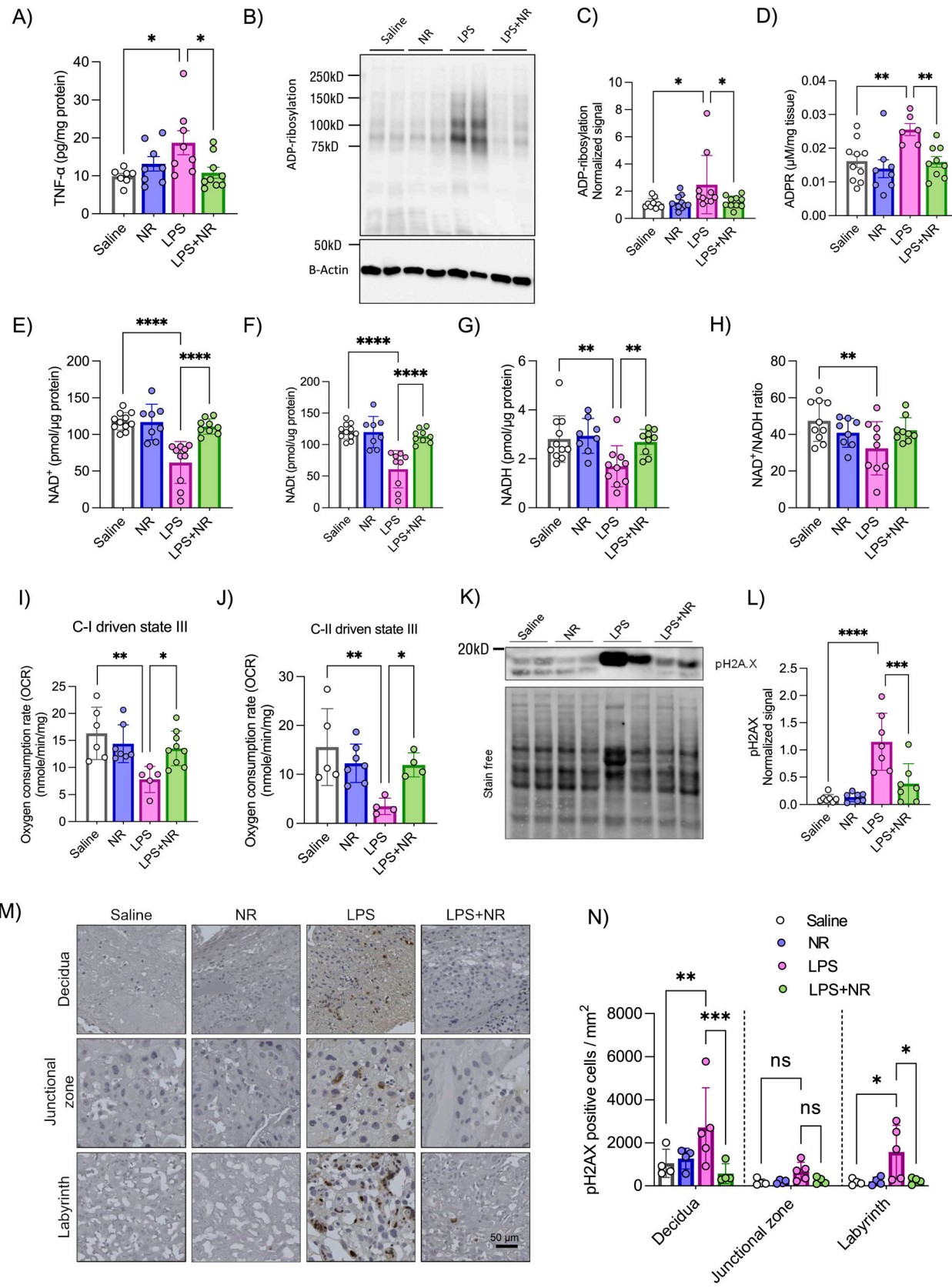

for 7 min and endogenous peroxidase blocking using 3% $H_2O_2$ in PBS for 30 min. To prevent non-specific binding, protein block solution (X0909; DAKO) was used for 1 h. Subsequently, the tissue sections were incubated overnight at 4°C with anti-ADP-ribosylation antibody (1:250, 83732; Cell Signaling) diluted in PBS containing 1% BSA, overnight at 4°C. Then, slides were incubated with Alexa Fluor 595 goat anti-rabbit secondary antibody (1:4,000, A-11012; Invitrogen) and kept in the dark for 1 h at RT. The tissue was counterstained with DAPI (P36931; Promega). Five snapshots of random regions of interest (20X magnification) of the placental villi (trophoblast) were selected for fluorescence intensity quantification using ImageJ software.

### Measurement of NAD(H)

Flash-frozen human placental samples (same cases as described above) were pulverized in liquid nitrogen to generate powdered tissue used for NAD(H) and mitochondrial protein analysis (see the Measurement of mitochondrial proteins section below). For measurements of NAD(H), 13–30 mg of powdered placental tissue was weighed in a 1.5-ml microcentrifuge tube and 1,000 µl of solvent A (40:40:20 acetonitrile:methanol:water with 0.1 M formic acid) was added. The mixture was vortexed for 10 s and kept on ice for 3 min. 87 µl of solvent B (15% $NH_4HCO_3$ in water) was added, vortexed, and kept on dry ice for 20 min to neutralize. The sample was centrifuged at 16,000$g$ for 15 min at 4°C, and the supernatant was collected for LC-MS analysis (91).

NAD, NADH, and NAM were determined using validated liquid chromatography–mass spectrometry/mass spectrometry (LC/MS/MS). After sample extraction, three internal standards (IS, 13C5NAD, 13C 2,6,7 NAM, and 6,7-dimethyl-2,3-di(2-pyridyl)-quinoxaline [DMDPQ]) were added to all samples (blanks, standards, quality controls [QC], and tissue extracts). After vortexing and centrifugation, the supernatant was transferred to autosampler vials for injection. The chromatographic separation was carried on an LC system (Accela, Thermo Fisher Scientific). The column was a Synergi Polar-RP (5 cm × 4.6 mm, 2.5 µm) column (Phenomenex). Separation was performed at 30°C with a gradient elution of acetonitrile/0.1% (vol/vol) formic acid in water at a flow rate of 0.5 ml/min. The autosampler was set at 4°C, and the injection volume was 10 µl. Mass spectral analyses were accomplished on a quadrupole tandem mass spectrometer (TSQ Quantum Access MAX; Thermo Fisher Scientific) with electrospray ionization in the positive mode. The multiple-reaction-monitoring mode was used to detect all compounds. Multiple-reaction-monitoring transitions were m/z 646 to 136, 666 to 649, 123 to 80, 669 to 136, 127 to 83, and 313 to 246 for NAD, NADH, NAM, 13C5NAD, 13C 2,6,7 NAM, and DMDPQ, respectively. The

calibration ranges were 200–20,000 ng/ml, 100–10 000 ng/ml, and 5–500 ng/ml for NAD, NADH, and NAM, respectively. The intraday and interday precision were all between 80 and 120%, and all extracted samples were stable at 4° for 24 h.

### Measurement of mitochondrial proteins

For measurements of mitochondrial proteins, 12.5 mg of powdered placental sample was added to 0.75 ml of cold lysis buffer (1% SDS in 50 mM triethylammonium bicarbonate [TEAB], Product No. 90114; Thermo Fisher Scientific), and homogenized by passing 20 times through 21G needle. The lysate was centrifuged at 20,000$g$ for 10 min at 4°C. The supernatant was separated, protein concentration was measured using BCA Protein Assay Kit (Product No. 23227; Thermo Fisher Scientific), and 100 µg of the sample was transferred into a new tube and adjusted to a final volume of 100 µl with 100 mM TEAB. 10 mM reduction buffer (from freshly prepared 1M dithiothreitol in water) was added and kept at RT for 1 h. 20 mM alkylation buffer (from 1 M iodoacetamide in water) was added and kept at RT for 45 min in the dark. Six volumes of pre-chilled protein precipitation buffer (50% acetone + 50% ethanol + 0.1% acetic acid) were added, vortexed, and allowed to precipitate overnight at −20°C. The next day, samples were centrifuged in a microfuge for 10 min at 10,000$g$. The supernatant was discarded carefully, and the remaining precipitation buffer was evaporated for 20 min at RT without over-drying. 100 µg of precipitated protein pellets was resuspended with 100 µl of 100 mM TEAB. Trypsin (Trypsin Gold; Promega) was used to digest the proteins (1:100 trypsin to protein by weight) and kept at 37°C overnight. Peptide labeling was performed using TMT10plex Isobaric Label Reagent Set plus TMT11-131C (A34808; Thermo Fisher Scientific), following the manufacturer's instructions.

After reconstitution in 20 µl 2% acetonitrile and 0.1% formic acid in water, 2 µl was injected for analysis. The system used consisted of UltiMate 3000 coupled with an Exploris 480 mass spectrometer (Thermo Fisher Scientific) equipped with a Nanospray Flex interface operated in a positive ion mode. The mobile phases consisted of 0.1% (vol/vol) FA in water as buffer A and 0.1% (vol/vol) FA in 80% acetonitrile as buffer B. The analysis was done on a column of 75 µm × 200 mm, packed also in-house with reverse-phase Magic C18AQ resins (3 µm; 120 Å pore size; Dr. Maisch GmbH).

Briefly, the sample was loaded on the column using 98% buffer A at a flow rate of 1 µl/min for 10 min. Then, a gradient from 5% to 35% buffer B was performed in 60 min at a flow rate of 300 nl/min. The MS method was set up following the TMT MS2 template in Xcalibur 4.3.73.11. Specifically, it consists of one full MS scan from 350 to 1,200 m/z, with a resolution of 120,000, defined at m/z 200 m, followed by

**Figure 4.   Effect of NAD⁺ boosting on placental dysfunction in a rat model of inflammatory PE.**
**(A)** Placental TNF-α levels were determined by ELISA (n = 7–9 litters/treatment, 1 placenta/litter). **(B, C)** Placental protein ADP-ribosylation was determined by Western blot with anti-ADP-ribosylation antibody (n = 10 litters/treatment, 1 placenta/litter). **(D)** Placental ADP-ribose (ADPR) levels were determined by LC/MS (n = 6–10 litters/treatment, 1 placenta/litter). **(E, F, G, H)** Placental NAD⁺(H) levels were determined by a colorimetric assay (n = 8–10 litters/treatment, 1 placenta/litter). **(I, J)** Placental mitochondrial function was determined by performing an oxygraph assay on isolated mitochondria. Complex I– and complex II–mediated oxygen consumption rates were recorded to determine their activity (n = 4–8 litters/treatment, 6–8 placentas/litter). **(K, L)** Placental oxidative stress was determined by Western blot with anti-pH2A.X antibody, a marker of oxidative DNA damage (n = 7 litters/treatment, 1 placenta/litter). **(M, N)** Placental sections were stained with anti-pH2A.X antibody to determine the number of pH2A.X-positive cells in different regions of placenta (n = 5 litters/treatment, 1 placenta/litter). Two-way ANOVA with Holm–Šídák's multiple comparisons test, *$P$ < 0.05, **$P$ < 0.01, ***$P$ < 0.001. The error bar indicates the SD. ADPR = ADP-ribose, LPS = lipopolysaccharide, NR = nicotinamide riboside.

the data-dependent MS/MS scan within 2 s, with dynamic exclusion of 1, exclusion duration of 45 s, and mass tolerance of 10 ppm, at an intensity threshold of 5E3. The isolation window was set to 0.7 D, and the HCD collision energy was 36. The resolution of MS2 was set to 45,000, with the first mass as 110 to include all the TMT tags. To improve the mass accuracy, all the measurements in the Orbitrap mass analyzer were performed with internal recalibration ("Lock Mass" at 445.120025). On the Orbitrap, the charge state rejection function was enabled, with only 2~5 charges included. The raw data were processed and analyzed using MaxQuant (version 1.6.6.0). A target–decoy database search was carried out for *Rattus norvegicus* UniProtKB/Swiss-Prot and UniProtKB/TrEMBL protein sequence databases including commonly observed contaminants and their reversed sequences. A precursor ion mass tolerance of 0.5 D was used to carry out this database search. TMT reporter intensities were corrected using the error correction factors provided with the kit.

### Measurement of oxidative stress

Placental tissue sections (5 $\mu$m) were deparaffinized and subjected to antigen retrieval as mentioned in the Measurement of ADP-ribosylation in Human sample processing section. DNA was denatured by treating the slides with 2 N HCl for 5 min at RT, followed by a neutralization step in 1M Tris base for 5 min at RT. Non-specific binding sites were blocked by incubating the sections in 10% normal goat serum in PBS, 1 h at RT. Next, tissues were incubated with 8-oxo-dG antibody (1:250, 4354-MC-050; R&D Systems) diluted in PBS containing 0.1% BSA, overnight at 4°C. Slides were incubated with Alexa Fluor 488 goat anti-mouse secondary antibody (1:4,000, A11029; Invitrogen) in PBS containing 0.1% BSA in the dark for 1 h at RT. The tissue was counterstained with 7-AAD (1:50, A1310; Invitrogen) diluted in water for 30 min at RT in the dark. Quantification was performed as described in the Measurement of ADP-ribosylation in Human sample processing section.

### Cell culture measurements

The HTR-8/SVneo cell line, derived from the human first-trimester placental villus explant, was purchased from the ATCC (CRL-3271). Cells were cultured in Gibco 1X RPMI 1640 (Life Technologies) medium supplemented with 10% FBS, at 37°C, 5% $CO_2$, and ~20% $O_2$ condition. Briefly, cells were treated with TNF-α (10 ng/ml) to induce inflammation and NR (250 $\mu$M) was co-treated for specified timepoints to test its potential to improve trophoblast mitochondrial and cellular function.

### Measurement of ADP-ribosylation

50,000 cells were plated on a 12-well plate and allowed to grow overnight. The next day, the cells were treated with TNF-α (10 ng/ml), with and without NR (150 $\mu$M–1 mM), for 24 h. A supplemented radioimmunoprecipitation assay (RIPA) buffer was prepared, which included Roche cOmplete Protease Inhibitor Cocktail (05892970001; Roche Canada), Roche PhosSTOP Phosphatase Inhibitor Cocktail (4906837001; Roche Canada), 5 mM of sodium butyrate, 100 $\mu$M of fresh tannic acid, and 1 $\mu$M of olaparib. Cells were then washed with ice-cold PBS, and RIPA lysis buffer was added to each dish. The

lysate was collected by scraping with a cell scraper (Thermo Fisher Scientific). The lysate was vortexed and kept in a rotor for 30 min for slow mixing. The lysate was then centrifuged at 20,000 relative centrifugal force (rcf) at 4°C for 15 min, and the supernatant was collected.

The protein concentration was determined using the DC protein assay (Bio-Rad Laboratories), with BSA standards prepared in RIPA buffer. The resulting supernatant was then transferred to new tubes and stored in a –80°C freezer. The samples were prepared with 4x Laemmli buffer (Cat# 1610737; Bio-Rad Laboratories) augmented with 10% β-mercaptoethanol (BP176-100; Fisher Bioreagents), and boiled for 5 min.

Next, Western blotting was performed using 8%, 10%, or 12% SDS–PAGE, depending on the size of the protein of interest. TGX Stain-Free FastCast Acrylamide Kit (Cat# 1610183; Bio-Rad Laboratories) with 10% ammonium persulfate was used to make the gel. After SDS–PAGE, the Stain-Free gels were activated, and then, the proteins were transferred to a Trans-Blot Turbo Mini size 0.2 µm nitrocellulose or PVDF membrane (Bio-Rad) using Trans-Blot Turbo Transfer System (Cat# 1704150EDU; Bio-Rad Laboratories). To detect the total protein, the blot was imaged again under a Stain-Free blot setting with automatic exposure time. Then, the blot was blocked for 1 h in blocking buffer (5% wt/vol BSA [SKU A7906; Sigma-Aldrich] in TBS-T buffer [50 mM Tris–HCl, pH 7.6; 150 mM NaCl; and 0.1% Tween]) while rocking gently at RT. The membranes were then incubated with primary anti-ADP-ribosylation antibody (AM80-100UG; MilliporeSigma) diluted in TBS-T buffer overnight at 4°C with gentle rocking. After three washes each for 10 min with TBS-T, the membrane was incubated for 1 h with the matching IgG, HRP-conjugated secondary antibody, rocking at RT. After three 10-min washes in TBS-T, detection was carried out on a ChemiDoc system (Bio-Rad Laboratories) using 1 ml Clarity (Bio-Rad Laboratories) or Clarity Max (Bio-Rad Laboratories) enhanced chemiluminescent solutions. Later, images were retrieved, and the quantification of blots was performed using either ImageLab or FIJI software.

### Measurement of NAD(H)

250,000 HTR-8/SVneo cells were plated on a 6-cm$^3$ plate and incubated overnight in serum-starved RPMI 1640 medium (Gibco 1X RPMI 1640; Life Technologies) at 37°C, 5% $CO_2$, and ~20% $O_2$ condition. The next day, 10% FBS containing RPMI 1640 medium was added, and cells were treated with 10 ng/ml TNF-α, with and without NR 250 $\mu$M, and incubated for 24 h. NAD(H) levels were quantified using a BioVision NAD$^+$/NADH kit (K337) according to the manufacturer's instructions.

### Measurement of mitochondrial respiration and the protein content

Mitochondrial respiration or OCR was measured by performing an XFe96 Seahorse assay. 2,500 cells/well were plated in a Seahorse cell culture plate in 180 $\mu$l of RPMI 1640 media with 10% FBS and incubated at 37°C, 5% $CO_2$ for 24 h. 12 h before running the assay, the cartridge was hydrated. 200 $\mu$l of XF Calibrant (pH 7.4) was added to each well of the calibration plate, and the cartridge was placed on the machine and sealed. The cartridge and calibration plate were placed in a 37°C incubator (no $CO_2$). 100 ml of Seahorse media was prepared

freshly before starting the assay. Medium constituents were as follows: 8.3 g/l DMEM powder (no H2CO3), 2 g/l D-glucose, 1 mM pyruvate, and 0.3 g/l glutamine, and made in 90 ml of sterile molecular grade water. The pH was set at 7.4, and the media were topped up to 100 ml with molecular grade water and filter-sterilized. Cells were washed with Seahorse media 2x. The Seahorse plate was placed in the 37°C incubator (no $CO_2$) for 45 min to allow the plate to equilibrate. All Seahorse drugs were prepared in Seahorse media and loaded in the cartridge. Calibration was performed, followed by measurements of the basal respiration. Oligomycin (1 $\mu$M) was injected to inhibit mitochondrial complex V to record non-phosphorylating OCR. FCCP (0.5 $\mu$M), a mitochondrial uncoupler, was injected to record maximal OCR. Antimycin A (1 $\mu$M) was injected to determine non-mitochondrial respiration to subtract from the recorded basal and maximal respiration. Monensin (200 $\mu$M) was injected to record the glycolytic capacity of the cells by recording extracellular acidification rates. Results were normalized by measuring the protein concentration of each well via the Bradford assay. Briefly, Seahorse media were removed from the plate using a multichannel pipette and cells were washed with 100 $\mu$l PBS. 30 $\mu$l of the lysis buffer (1 mM Tris–HCL, pH 7.4, 1 mM EDTA, and 0.5% Triton X-100) was added to each well using a multichannel pipette and mixed gently, with 5 $\mu$l from each well mixed with 195 $\mu$l of 1:5 Bradford reagent (Bio-Rad) for protein quantification. Absorbance was read at 595 nm using a spectrophotometer. The measurement was compared with BSA protein standards to get relative quantification.

Mitochondrial OXPHOS protein expression was assessed by Western blotting. Briefly, HTR-8/SVneo cells were cultured as per conditions described in the Measurement of ADP-ribosylation in Cell culture measurements section (TNF-$\alpha$ [10 mg/ml] ± NR [250 $\mu$M] for 24 h). The Western blotting protocol was similar to that described in the Measurement of ADP-ribosylation in Cell culture measurements section, using the OXPHOS antibody cocktail (ab110413).

### Measurement of invasive capacity

250,000 HTR8/SVneo cells were plated on 6-$cm^3$ plates. After overnight incubation, cells were treated with TNF-α (10 ng/ml) ± NR (250 $\mu$M) in 10% FBS containing RPMI 1640 media for 48 h. Media were changed at 24 h, with replenishment of TNF-$\alpha$ (and/or NR) treatment. Matrigel-coated invasion inserts were used in a Boyden chamber invasion assay. Briefly, a 24-well plate containing Matrigel inserts was removed from the freezer and warmed to RT (30 min). 750 $\mu$l of serum-free warm RPMI 1640 media was added to the bottom of each well, with an additional 500 $\mu$l media added on top of each insert, and incubated at 37°C for 2 h to rehydrate the Matrigel. HTR8/SVneo cells were then trypsinized, neutralized, and transferred to the top portion of the chamber onto the Matrigel (50,000 cells/well) in serum-depleted RPMI 1640 media. Serum-enriched media were placed in the bottom of the chamber, promoting cellular migration through the Matrigel. The cells were incubated in the chamber overnight (with all the treatments) at 37°C, at 5% $CO_2$ and ~20% $O_2$ condition.

After a 24-h incubation, media were removed from the top and bottom of the chamber, with the insert removed and washed with PBS (2x). Then, 700 $\mu$l of 10% formalin was added to the wells, fixing

the adherent cells to the insert at RT for 10 min. Inserts were removed from the fixative and washed 2x with PBS. Inserts were then placed in 100% methanol for 20 min at RT to permeabilize the cells. Inserts were then washed 2x with PBS, and adherent cells were stained with hematoxylin for 10 min at RT. Inserts were washed once more with PBS, followed by tap water twice. Using a cotton swab, any cells remaining in the top chamber (non-invaded cells) were removed. The insert was allowed to air-dry, after which the membrane was cut out and mounted on a glass slide.

### Animal model measurements

All animal experiments were carried out according to the University of Ottawa animal care ethics and guidelines (protocol# HS2923). Sprague Dawley rats (Charles River Laboratories International, Inc.) were kept in a room with 23°C temperature and a 12:12-h light–dark cycle. The estrous cycle was confirmed by doing vaginal smears to allow timed mating. Vaginal smears were checked the next day to identify the presence of sperm to confirm pregnancy. In addition, body weights were recorded daily to track pregnancy. The following experimental groups were included:

LPS (n = 12)—LPS (*Escherichia coli* O55:B5, L2880-10MG; Sigma-Aldrich) was solubilized in sterile normal saline, and animals were intraperitoneally (IP) injected from GD 13–18 at a daily incremental dose from 20 to 70 $\mu$g/kg/day, as shown previously ([92]).

NR (n = 10)—NR was administered by oral gavage from GD 1–19. NR was solubilized in drinking water and sterile-filtered and given at a concentration of 200 mg/kg/day.

LPS + NR model (n = 10)—NR and LPS were administered as mentioned above.

Saline (n = 10)—Controls received oral gavage of sterile drinking water from GD 1–19 and saline IP injections from GD 13–18.

Animals were euthanized on GD 19, fetal and placental weights and litter size were recorded, and organs were collected for future analysis.

### Measurements of pregnancy outcomes

Maternal mean arterial blood pressure (MAP) was assessed across the last portion of pregnancy (GD 14–19) for each of the four different treatment groups (control, NR alone, LPS alone, and LPS + NR; n = 7–8/group). Measurements were captured using the CODA high-throughput non-invasive tail–cuff measurement system (Kent Scientific Corporation). Animals were first acclimatized to the system for 4–5 d. Before recording, animals were stabilized for 5–10 min, and body temperature was checked. Then, BP was recorded for 20 min on each testing day.

At GD 19, animals were euthanized. The litter size, number of resorptions, fetal weights, and placental weights were collected. Placentas were collected and flash-frozen for RNA, metabolite, and protein measurements. One placenta per litter was fixed in 4% PFA for histopathology.

### Measurement of placenta histomorphology

PFA-fixed midline placental tissues (n = 1 placenta/litter; n = 10 litters/group) were paraffin-embedded, sectioned at 5-$\mu$M thickness, and placed on glass slides. Sections were deparaffinized, rehydrated, and stained with H&E using standard protocols ([93]). Whole-slide

images were captured using an Axio Scan.Z1 microscope at 20X magnification. Measurements of the total placenta midline cross-sectional area, as well as cross-sectional areas of each of the individual regions of the placenta (labyrinth zone, junctional zone, and decidua), were analyzed using ZEN 3.1 blue edition.

### Measurement of placenta gene expression

RNA was isolated from frozen placental tissues (n = 1 placenta/litter; n = 10 litters/group) using standard TRIzol methods. Purified RNA was submitted to StemCore Laboratories at the Ottawa Hospital Research Institute for RNA sequencing as follows. RNA quantification and quality were assessed using a Qubit HS RNA assay (Q32852; Thermo Fisher Scientific) and the Fragment Analyzer Standard Sensitivity RNA assay (DNF-471-0500; Agilent). All samples had an RNA quality number above 8.0. The DNA library was prepared using Illumina Stranded mRNA Preparation Kit (Illumina). Libraries were quantified using the Qubit Double-Stranded DNA HS kit (Q33230; Thermo Fisher Scientific) and integrity-confirmed with the high-sensitivity NGS assay on AATI Fragment Analyzer (Agilent). DNA libraries were normalized, pooled, and diluted. 75-base pair single-end sequencing reads were obtained from pooled DNA libraries using the Illumina NS500 75-cycle high output kit on Illumina NextSeq 500 Sequencer.

Sequencing results were separated by library barcoding, and barcodes were removed using Illumina BaseSpace. Subsequently, data were imported into the Galaxy platform maintained by GenAP (http://www.genap.ca) and hosted by the Digital Research Alliance of Canada. The quality of sequencing was assessed using the FastQC function. Sequences were aligned to the rat genome (mRatBN7.2), and read quantification was performed using the HISAT2 and featureCounts functions, respectively. Count normalization, differential gene expression, and principal component analyses between the control and LPS-treated group were performed using DESeq2 and an FDR of 10%. Using only the differentially expressed genes with LPS treatment, to determine genes rescued with NR treatment, a subsequent analysis using DESeq2-identified genes that changed between LPS and LPS with NR groups with an FDR of 10% was performed.

Z-scores were calculated from normalized counts of differentially expressed genes and heatmaps generated in RStudio. Gene set enrichment analysis was performed using WebGestalt (http://www.webgestalt.org/) and the Reactome and Gene Ontology Biological Process databases.

### Measurement of the placenta TNF-α protein content

Placenta TNF-$\alpha$ protein expression was measured using the Quantikine Rat TNF-$\alpha$ Immunoassay immunosorbent assay (R&D Systems). The frozen placental tissue (n = 1 placenta/litter; n = 10 litters/group) was pounded in liquid $N_2$ and weighed about 40 mg for homogenizing 5 mM EDTA/PBS extraction buffer with Protease Inhibitor Cocktail (11836170001; cOmplete Roche). TNF-α ELISA was then carried out according to the manufacturer's protocol.

### Measurement of protein ADP-ribosylation

The flash-frozen placenta (n = 1 placenta/litter; n = 10 litters/group) was pulverized, and 15–25 mg of tissue was used. RIPA buffer was added to the tissue and homogenized using a 21G syringe. The homogenate was vortexed for 30 min to ensure complete

homogenization. The homogenate was centrifuged at 20,000 relative centrifugal force (rcf) for 15 min at 4°C, and the supernatant was transferred to a 1.5-ml microcentrifuge tube. Western blotting was performed as described in the Measurement of ADP-ribosylation in Cell culture measurements section using the anti-ADP-ribosylation antibody (AM80-100UG; MilliporeSigma).

### Measurement of NAD(H)

30–40 mg of powdered placental tissue (n = 1 placenta/litter; n = 10 litters/group) was added to 300 $\mu l$ of extraction buffer. To prepare a homogeneous solution, the solution was drawn up and down through a 21-gauge syringe needle. Quantification was performed using the BioVision $NAD^+/NADH$ kit (K337), according to the manufacturer's instructions.

### Measurement of NAM and ADPR

Placental ADPR and NAM levels were quantified at the metabolomics core facility of the University of Ottawa. Approximately 40 mg (±10%) of powdered placental tissue (generated from n = 1 placenta/litter; n = 10 litters/group) was taken and added to 1.8 ml pre-chilled (–80°C) 80% methanol. Four 2.8-mm ceramic beads were added to each tube, and the samples were homogenized at 324 rcf for 45 s using a MagNa Lyser bead homogenizer for up to three cycles. To keep samples cool between cycles, the tubes were kept on dry ice for 1 min. After homogenization, 1.8 ml of pre-chilled dichloromethane (–20°C) and 900 μl of ice-cold molecular grade water were added to the homogenate. For lipid removal, the mixture was vortexed and kept on ice for 10 min for partitioning. Samples were then centrifuged at 1,252 rcf at 1°C for 10 min, and the upper aqueous phase was carefully collected and dried at 4°C using a refrigerated centrifugal concentrator (Labconco Refrigerated CentriVap Benchtop Vacuum Concentrator).

For quantification, 0.5 $\mu g/ml$ of deuterated d5-Trp (Tryptophan) and 0.05 $\mu g/ml$ of d4-NAM were used as internal standard (IS) in 75% acetonitrile to reconstitute the study samples. The mix was further tested in samples to check the carryover or matrix effect on the endogenous metabolite level. Samples were reconstituted to a final volume of 50 $\mu l$. Samples were used undiluted for ADPR measurements, and 10X diluted (in 75% acetonitrile) for NAM measurements. Mass spectral analyses were accomplished on a quadrupole time-of-flight mass spectrometer (MS) (6545B; Agilent), with electrospray ionization in a positive mode connected to an ultra-high-performance liquid chromatography system (1290 Infinity II; Agilent). 2 $\mu l$ of samples was injected into the system, and Agilent Q-TOF Quantitative Analysis software (Agilent Technologies Inc.) was used to process and analyze the data. The calibration curve was optimized for each metabolite. After peak detection, the peak area was integrated using the Agile2 integrator. A >10 signal/noise ratio and an accuracy of 80–120% were used. Metabolite concentrations were normalized by the IS signal and adjusted to account for dilution factors. Metabolite concentrations were then normalized by the tissue weight.

### Measurement of rat placental mitochondrial respiration and the protein content

Freshly isolated mitochondria were used to measure mitochondrial respiration rates using a high-resolution oxygraph assay. The fresh

placental tissue (n = 6–8 placentas/litter; n = 10 litters/group) was collected to isolate mitochondria according to a modified version of a previously published protocol (94). Briefly, placentas were washed in ice-cold PBS and transferred to a beaker containing buffer A (300 mM sucrose mM, 10 mM Tris–HCl, 1 mM EGTA/Tris base, and 0.1% BSA [fatty acid-free], pH 7,2, kept on ice). Placentas were cut into smaller pieces and washed with buffer A twice to eliminate residual blood. Placentas were minced and transferred to a Potter–Elvehjem homogenizer (30 ml) and homogenized at 500 rpm for six strokes. The homogenate was centrifuged for 10 min at 1,000rcf at 4°C. The supernatant was transferred to a clean tube and centrifuged again with the same settings. Next, the supernatant was collected using a syringe to eliminate the fat layer. The supernatant was collected in a clean tube and centrifuged for 10 min at 8,000rcf at 4°C. The supernatant was removed, and the pellet was resuspended in 2 ml of buffer B (300 mM sucrose, 10 mM Tris–HCl, and 0.05 mM EGTA/Tris base, pH 7) and centrifuged for 10 min at 8,000$g$ at 4°C. This step was repeated. 200 $\mu$l of MiR05 respiration buffer (110 mM D-sucrose, 60 mM lactobionic acid, 20 mM taurine, 20 mM Hepes, 10 mM KH2PO4, 3 mM MgCl2, 0.5 mM EGTA, and 1 g/l fatty acid–free BSA, pH 7.1, at 23°C) (95, 96) was added to the pellet and gently mixed to make a mitochondrial suspension. The mitochondrial protein content was measured by performing a detergent-compatible colorimetric (DC) protein assay (Bio-Rad).

Mitochondrial respiration rates were determined by an oxygraph assay, according to a published protocol (97). Briefly, mitochondria (2 mg/ml in MiR05 respiration buffer) were added to the chamber and baseline traces were recorded. Substrates and inhibitors of mitochondrial respiration were injected in the following series: (i) 5 mM glutamate and 2.5 mM malate, a complex I substrate; (ii) 1 mM ADP, to measure complex I–driven state-III respiration; (iii) 1 mM amytal to inhibit complex I; (iv) 5 mM succinate to measure complex II–driven state-III respiration; (v) 5 $\mu$M antimycin A to inhibit complex III to prevent electron transfer to complex IV. This allows to measure complex IV–mediated respiration by adding substrate exogenously; (vi) 5/0.3 mM tetramethyl-p-phenylenediamine/ascorbate (TMPD)/ascorbate to measure complex IV–driven state-III respiration; and (vii) finally, potassium cyanide (KCN) to inhibit complex IV.

Mitochondrial protein expression was measured in flash-frozen placental tissues (n = 1 placenta/litter; n = 10 litters/group). Samples were prepared as described inthe Measurement of protein ADP-ribosylation section, and Western blotting was performed according to the methods described in the Measurement of ADP-ribosylation in Cell culture measurements section, using anti-OXPHOS (ab110413; Abcam), anti-OPA1 (ab42364; Abcam), YME1L1 (11510-1-AP; Proteintech), and anti-CLPP (ab124822; Abcam) antibodies.

#### Measurement of rat placental oxidative stress

Measurements of placental pH2A.X protein expression, a marker of double-stranded DNA breaks and oxidative stress, were assessed using frozen placental tissues (n = 1 placenta/litter; n = 10 litters/group) by Western blot. Samples were prepared as described in the Measurement of placenta TNF-$\alpha$ protein content section, and Western blotting was performed according to the Measurement of ADP-ribosylation in Cell culture measurements section. using the pH2A.X antibody (#2577; Cell Signaling). In parallel, paraffin-embedded placental tissue sections from the same litters (n = 1 placenta/litter; n = 5 litters/group) underwent deparaffinized, antigen retrieval and blocking for non-specific binding as described in the Measurement of oxidative stres section. Tissue sections were incubated overnight at 4°C with the pH2A.X antibody (1:300, #2577; Cell Signaling) diluted in PBS containing 1% BSA. Slides were then incubated with a secondary antibody for 1 h at RT, followed by incubation with streptavidin–HRP for 30 min, using the LSAB2 system–HRP (K0675; DAKO). The reaction was developed using 3,3'-DAB (D5637; Sigma-Aldrich) plus H2O2. The tissues were counterstained with the Harris hematoxylin and mounted in Entellan. Whole-slide images were captured using a Zeiss Axio Scan Z.1 scanner at 20X magnification and analyzed using QuPath software (98). The decidua, junctional zone, and labyrinth were delimited to perform layer-specific automated positive cell counting. The results were expressed as pH2A.X-positive cells per square millimeter.

### Statistical analysis

Statistical analysis was performed using GraphPad Prism software (version 9.5). Error bars in graphical presentations indicate the mean ± SD. Depending on the treatment groups, an unpaired $t$ test or one-way or two-way ANOVA with a Holm–Šídák multiple comparisons test was applied. For animal experimentations, the robust regression and outlier removal (ROUT) method (available on GraphPad Prism) was applied to exclude any outliers. A $P$-value < 0.05 was considered statistically significant.

## Supplementary Information

## Acknowledgements

This research was funded by the Canadian Institutes of Health Research (CIHR) Project Grant #PJT-153055 to SA Bainbridge and KJ Menzies. F Jahan is supported by Frederick Banting and Charles Best Canada Graduate Scholarship from CIHR (FRN-167027).

### Author Contributions

F Jahan: conceptualization, data curation, formal analysis, validation, investigation, visualization, methodology, and writing—original draft and project administration.
G Vasam: formal analysis, validation, investigation, visualization, methodology, and writing—review and editing.
Y Cariaco: data curation, formal analysis, investigation, visualization, methodology, and writing—review and editing.
A Nik-Akhtar: formal analysis, visualization, and writing—review and editing.
A Green: data curation, formal analysis, investigation, visualization, and methodology.

KJ Menzies: conceptualization, resources, supervision, funding acquisition, investigation, methodology, project administration, and writing—review and editing.
SA Bainbridge: conceptualization, resources, supervision, funding acquisition, investigation, methodology, project administration, and writing—review and editing.

## Conflict of Interest Statement

The authors declare that they have no conflict of interest.

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
