## [Reviewer comments · Life Science Alliance]

Life Science Alliance

NAD⁺ depletion is central to placental dysfunction in an inflammatory subclass of preeclampsia

Fahmida Jahan, Goutham Vasam, Yusmaris Cariaco, Abolfazl Nik Akhtar, Alex Green, Keir Menzies, and Shannon Bainbridge
DOI: <https://doi.org/10.26508/lsa.202302505>

Corresponding author(s): Shannon Bainbridge, University of Ottawa and Keir Menzies,

Review Timeline:	Submission Date:	2023-11-30
	Editorial Decision:	2024-01-26
	Revision Received:	2024-08-07
	Editorial Decision:	2024-09-04
	Revision Received:	2024-09-23
	Accepted:	2024-09-24

Transaction Report:

January 26, 2024

Re: Life Science Alliance manuscript #LSA-2023-02505-T

Dr. Shannon A Bainbridge
University of Ottawa
Interdisciplinary School of Health Sciences
CANADA

Dear Dr. Bainbridge,

Thank you for submitting your manuscript entitled "NAD+ depletion is central to placental dysfunction in an inflammatory subclass of preeclampsia" to Life Science Alliance. The manuscript was assessed by expert reviewers, whose comments are appended to this letter. We invite you to submit a revised manuscript addressing the Reviewer comments.

Thank you for this interesting contribution to Life Science Alliance. We are looking forward to receiving your revised manuscript.

Sincerely,

B. MANUSCRIPT ORGANIZATION AND FORMATTING:

Reviewer #1 (Comments to the Authors (Required)):

Jahan and coworkers have an interesting and original paper showing that NAD⁺ coenzyme depletion occurs within and appears to mediate aspects of an inflammatory class of preeclampsia. The best parts of the paper are figures 2 through 4. The worst parts of the paper are figure 1, aspects of the referencing and overall writing. I will try to guide the authors to a better paper because the data deserve to be published in LSA.

1. Suggest they change "singular disease" to unique molecular or pathological entity.
2. The sentence that begins in line 99 and ends in 102, citing 13 articles, is really not good. Those papers do not specifically implicate mitochondria and they include the good, the bad and the ugly. They don't even include the most relevant primary literature on transcriptional activation of MARYlating PARPs in inflammatory conditions, which was done by Heer et al (JBC and then reviewed in Nature Met with some translational citations). If you have conditions that are disturbing NAD, you can disturb mitochondrial functions, anabolic functions, DNA repair functions, calcium functions, etc. The authors are urged to take time to read the key papers and cite the key primary papers rather than reviews that regurgitate poorly tested ideas. There is a place for reviews but this is not it.
3. If the authors want to bring in Sirtuins, they should read Sirtuins are not conserved longevity genes by Brenner in Life Metabolism. I don't know if there is really a common theme of what sirtuins do any more than what the PARP superfamily does.
4. Though the treatment of the PARP superfamily in the discussion is better, it's really poor in the introduction. It is not true that most members of the PARP superfamily make PAR or have much to do with DNA repair.
5. Reference needed for nicotinamide riboside (Bieganowski & Brenner, 2004).
6. I don't think Figure 1 makes sense because one has to use a magnifying glass to see that the overexpressed genes are PARP4, PARP9, PARP13 and PARP14 (it actually looks like figures 1 and 2 from Heer et al). It doesn't say in the text that those are the PARP superfamily members that are induced but that's apparently the result. Figure 1B then goes on to probe for PAR, which does not follow from Figure 1A. Based on the DAPI images, there's better nuclear staining in PE2 and PE3, so it is not surprising that there is more PAR staining in PE2 and PE3. I don't really want to see those data "quantified" in Fig 1C. It does not look meaningful. NAD data in Figure 1D support the general idea of the figure but the authors looking for PAR rather than either total MARYlation or MARYlated PARP4,9,13,14 is not a good way to begin this paper.
7. Figures 2-4 were pretty nice. Figure 2 in this paper was also reminiscent of Heer.
8. On line 291, there's an "absence of evidence" problem in discussion of CD38. PARP1 isn't transcriptionally induced either so the authors already have a problem if they want transcriptional changes to line up with what is the major consumer of NAD⁺.
9. While the paragraph beginning on line 338 is good, this underscores the problem with Figure 1.
10. Don't say "reduced NAD⁺" when you mean depressed NAD coenzyme levels. This was also discussed by Brenner in Life Metabolism.

Nicely complementary reviews. The authors should revise in response to both constructive reviews

Reviewer #2 (Comments to the Authors (Required)):

This paper is very complex. There are 3 experiments in one paper, one with human placenta, a cell culture study and a pregnant rat study. It might be worth considering just having the human data in a study by itself so the authors could expand more on those findings and what they used to divide the groups of PE.

The reviewer is having a little hard time discerning inflammation-driven PE (PE3) from ischemia-driven PE (PE2), is this the hypoxia driven group and which markers differentiate them?

The control group should not consist of preterm patients, these should be in a group of their own. Moreover, control patients do not deliver at 34 weeks gestation, they are at least 37.

The reviewer is not sure I am seeing any bands on the blot except for Cox V (Fig 2H). there is a good separation. The B actin looks as though it is on a separate blot because of the migration of the lanes

The mean blood pressures are not high enough for a PE rat model, they could be for an infection (Fig 3B). Their blood pressure is only 100 over what looks like 90 for the saline groups. Please state these numbers in the results. Moreover, 60% of the rats with LPS died. This would indicate to the reviewer it is more of an infection during pregnancy model and the LPS dose is too high.

We hypothesized that there is a decline in placental NAD⁺ levels due to 411 hyperactivity of NAD⁺ consuming enzymes and that it leads to impaired placental 412 mitochondrial function, collectively contributing to the development of PE. Thus, 413 replenishing NAD⁺ could be an attractive preventative strategy in this subclass of PE 414 patients." This should be in the rationale for the study not the methods

"Placental mitochondrial dysfunction has long been recognized as a key component of PE pathophysiology [8-12]. Altered placental mitochondrial respiration and/or total content is observed in tissues collected from pregnancies that span the clinical spectrum of PE, associated with altered oxidative phosphorylation (OXPHOS) protein expression, mitochondrial structural damage, defects in fusion/fission dynamics and increased oxidative stress [8, 13-15]. There is strong evidence to suggest that placental mitochondrial dysfunction may be a common feature across all PE subclasses [8],"

placenta mitochondrial functionality assay and protein expression, and oxidative stress discussion should be in the methods not in the supplement.

I suggest authors discuss previous findings on TNF-alpha mediated mitochondrial damage in PE rat model (doi.org/10.1016/j.preghy.2021.02.006) it isn't in any of these references and it could support the studies with TNF.

10mg is really supraphysiological concentration of TNF alpha and is not what is seen in PE. The Results state 10 ng/ml which is different than methods

Some of the results repeat what is in the methods and could be reduced in the results section, especially with the rat model description.

The subtitles for the sections in results are way too long

1. An experiment demonstrating or a discussion on how reduced NAD⁺ levels would lead to reduced placental (or trophoblast) mitochondrial function is missing? Do authors believe SIRT6 are involved?
2. The link between PARP (DNA repair enzyme localized to nucleus) and mitochondrial function modulation via NAD could be better discussed.
3. It is understandable authors chose TNF-alpha based on their previous microarray work, do you believe PARP activation is TNF-alpha mediated entirely or other inflammatory mediators recapitulate these findings?
4. For example, a published study (doi.org/10.1091/mbc.E18-10-0650) shows PARP1 inhibits hexokinase (glycolysis), can you comment on your ECAR data in Fig 2G?
5. Page 8, Line 169: Several studies demonstrate reduced COX-IV expression in preeclamptic patients. Hence, it may be not a suitable marker for mitochondrial content.
6. Is there a reason you used different markers of DNA damage? 8-oxo-DG (human placenta) and p-H2A.X (rat placenta)? Measurement on mitochondrial oxidative damage would have been a good addition to establish mito damage in this study.
7. Other studies in human PE placentas suggest that mt ROS is down, but there is no discussion of these findings compared the present work
8. Page 33, Fig 2: Did you validate increase in NAD⁺ levels after treating trophoblasts with NR?
9. Page 33, Fig 2H: Is there a reason complex I was not probed? Especially it being NADH dehydrogenase. Did you measure individual electron transport chain activities
10. Page 37: Did you measure electron transport chain complex expression and activities in rat placenta? (Can you please check it in supplemental?)
11. is an impairment in mitochondria or the effects are just a reflection of metabolic shift.
12. Page 37, Fig 4E: Can you comment on why you didn't see increased NAD levels in NR vs. Saline group?
13. Can you comment on how PARP inhibitors would be an alternative approach to NAD⁺ replenishment?

We thank the reviewers for the comprehensive and helpful comments. We have taken the time to consider each provided comment and revise the manuscript accordingly. Specific responses to reviewer comments can be found below in red.

Shannon Bainbridge
On behalf of all authors

Reviewer #1:

Jahan and coworkers have an interesting and original paper showing that NAD⁺ coenzyme depletion occurs within and appears to mediate aspects of an inflammatory class of preeclampsia. The best parts of the paper are figures 2 through 4. The worst parts of the paper are figure 1, aspects of the referencing and overall writing. I will try to guide the authors to a better paper because the data deserve to be published in LSA.

1.1. Suggest they change "singular disease" to unique molecular or pathological entity. **This has now been changed in the introduction.**

1.2. The sentence that begins in line 99 and ends in 102, citing 13 articles, is really not good. Those papers do not specifically implicate mitochondria and they include the good, the bad and the ugly. They don't even include the most relevant primary literature on transcriptional activation of MARYlating PARPs in inflammatory conditions, which was done by Heer et al (JBC and then reviewed in Nature Met with some translational citations). If you have conditions that are disturbing NAD, you can disturb mitochondrial functions, anabolic functions, DNA repair functions, calcium functions, etc. The authors are urged to take time to read the key papers and cite the key primary papers rather than reviews that regurgitate poorly tested ideas. There is a place for reviews but this is not it.

We thank the reviewer for this suggestion. We have now updated the references with relevant papers.

1.3. If the authors want to bring in Sirtuins, they should read Sirtuins are not conserved longevity genes by Brenner in Life Metabolism. I don't know if there is really a common theme of what sirtuins do any more than what the PARP superfamily does.

We have now excluded sirtuins from the introduction as our paper did not pursue the role of sirtuins.

1.4. Though the treatment of the PARP superfamily in the discussion is better, it's really poor in the introduction. It is not true that most members of the PARP superfamily make PAR or have much to do with DNA repair.

We thank the reviewer for identifying this oversight. We have now made the following correction.

"Enzymes such as ADP-ribosyltransferases (ARTs), Sirtuins (SIRT), cluster of differentiation 38/157 (CD38/CD157), and sterile alpha and toll/interleukin-1 receptor motif-containing protein 1 (SARM1), require NAD⁺ as a co-substrate to carry out their

enzymatic functions [26-29]. Among these, the most well-described NAD^+ -consuming enzyme family is the ARTs, of which includes the diphtheria toxin-like family (ARTDs) that perform most of the ADP-ribosylation in mammals and are often referred to using their old family nomenclature of PARPs [30]. A primary function of PARPs is the post-translational modification of proteins, via the covalent addition of ADP-ribose polymer(s) to various proteins – a process known as mono-ADP-ribosylation (MAYylation) when a single unit of ADP-ribose is added or polyADP-ribosylation (PARylation) when multiple ADP-ribose units are added [30]. This NAD^+ -dependant process helps to regulate numerous biological processes such as chromatin organization, mRNA stability, transcriptional control, DNA methylation, glycolysis, inflammation, immune response and DNA damage repair [17, 30-36].”

1.5. Reference needed for nicotinamide riboside (Bieganowski & Brenner, 2004).

Now added.

1.6. I don't think Figure 1 makes sense because one has to use a magnifying glass to see that the overexpressed genes are PARP4, PARP9, PARP13 and PARP14 (it actually looks like figures 1 and 2 from Heer et al). It doesn't say in the text that those are the PARP superfamily members that are induced but that's apparently the result. Figure 1B then goes on to probe for PAR, which does not follow from Figure 1A. Based on the DAPI images, there's better nuclear staining in PE2 and PE3, so it is not surprising that there is more PAR staining in PE2 and PE3. I don't really want to see those data "quantified" in Fig 1C. It does not look meaningful. NAD data in Figure 1D support the general idea of the figure but the authors looking for PAR rather than either total MAYylation or MAYylated PARP4,9,13,14 is not a good way to begin this paper.

We appreciate the reviewer's concerns and suggestions and have tried to reconfigure figure 1 considering these. Since we did not investigate the roles of individual PARP superfamily proteins members, we have chosen to exclude the original PARP gene expressing panel (original Panel 1A). We have also organized the presented data collected in line with the proposed hypothesis being tested.

- We first establish the presence of a pro-inflammatory phenotype in the placentas of the PE3 subclass (Panel 1A).
- We next demonstrate a hyperactivation of ADP-ribosylation (Fig 1B-C) and reduced placental NAD^+ content (Fig 1D-G) in these same inflammation-mediated PE samples.
 - Importantly, the ADP-ribosylation staining presented was performed using a Poly/Mono-ADP Ribose (E6F6A) antibody from Cell Signalling. In the past version of the manuscript, we incorrectly labelled these results as PARylation alone. We have now corrected this error, adding “ADP-ribosylation” in the figure label.
 - We would also like to clarify that one of the main reasons for choosing immunofluorescence to detect ADP-ribosylation complexes, vs Western blot, was the ability to determine the spatial localization of these complexes within the chorionic villous exchange structure, allowing us to specifically determine if these complexes were enriched in the syncytiotrophoblast cellular layer which is

- in direct contact with the maternal blood and which fulfills pivotal endocrine and maternal-fetal exchange functions required to sustain fetal growth.
- In the revised figure, we have also prepared better representative images of the DAPI staining using pre-established image capture settings – allowing us to better demonstrate the observed stability of DAPI staining across samples both within and across PE subclasses. It is worth noting that human placenta often displays syncytial knots containing multiple condensed nuclei which exhibit bright DAPI fluorescence. In disease conditions such as PE, syncytial knots are more abundant in relation to healthy placentas (PMID: 17140657). Thus, in some areas it may appear to exist greater DAPI signal from such areas in PE groups, but this is a unique feature of these structures.
 - Next, we show the downregulation of several mitochondrial proteins (Fig 1H) and oxidative DNA damage (Fig 1I-J) in PE3 as mitochondrial dysfunction and oxidative damage are key features observed in PE when studied as one pathological disease.

1.7. On line 291, there's an "absence of evidence" problem in discussion of CD38. PARP1 isn't transcriptionally induced either so the authors already have a problem if they want transcriptional changes to line up with what is the major consumer of NAD+. We thank the reviewer for identifying this issue. As we did not determine the activity of CD38 enzyme in the placenta and cannot conclude its involvement in the presented context, we have now removed this data of protein expression.

1.8. While the paragraph beginning on line 338 is good, this underscores the problem with Figure 1. We have now made modifications in this paragraph according to the present changes to the figure 1.

1.9. Don't say "reduced NAD+" when you mean depressed NAD coenzyme levels. This was also discussed by Brenner in Life Metabolism. We have now made the corrections.

Reviewer #2:

2.1. This paper is very complex. There are 3 experiments in one paper, one with human placenta, a cell culture study and a pregnant rat study. It might be worth considering just having the human data in a study by itself so the authors could expand more on those findings and what they used to divide the groups of PE.

We appreciate this suggestion by the reviewer. However, we do strongly believe that the complementary nature of these distinct datasets is a strength of this body of work and have chosen to keep them together within the manuscript.

2.2. The reviewer is having a little hard time discerning inflammation-driven PE (PE3) from ischemia-driven PE (PE2), is this the hypoxia driven group and which markers differentiate them?

PE3 is differentiated based on its placenta inflammatory gene signature profile and placenta histological features showing heightened inflammation, as shown in figure 1A. PE2 is the hypoxia subclass or “canonical” preeclampsia characterized by overwhelming evidence of maternal vascular malperfusion and placental hypoxia. Markers for each of the 3 subclasses of PE, along with detailed phenotypic profiling of each subclass, has been described in detail in the reference papers from our group (Leavey et al 2016 and subsequent studies of Gibbs et al 2018 and Benton et al 2018).

2.3. The control group should not consist of preterm patients, these should be in a group of their own. Moreover, control patients do not deliver at 34 weeks gestation, they are at least 37.

We have combined the healthy term and preterm as control group since the PE group also consists of both term and preterm placentas. Due to a smaller number of patients in PE1 and PE3, we could not do a comparison of control term vs PE term and control preterm vs PE preterm. However, we did determine the difference between control term and control preterm for all the parameters (shown below). Except for NAD⁺(H) levels, we did not find any statistically significant difference between control term and preterm. For NAD⁺(H) levels, they are high in control preterm compared to term healthy placenta suggesting a gestational age- dependant decline in NAD⁺(H) levels. Therefore, it was important to combine control term and preterm as PE subclasses also included term and preterm patients.

2.4. The reviewer is not sure I am seeing any bands on the blot except for Cox V (Fig 2H). there is a good separation. The B actin looks as though it is on a separate blot because of the migration of the lanes

In figure 2H, Complex-IV and Complex-II bands were faint. We have attached the full blot with two different exposure times that were used for quantification purposes. For normalization, a stain free blot image (total protein loading) was used, and the full image has been attached to clearly demonstrate it is in fact from the same blot.

Full unedited blot for Figure 2H

2.5. The mean blood pressures are not high enough for a PE rat model, they could be for an infection (Fig 3B). Their blood pressure is only 100 over what looks like 90 for the saline groups. Please state these numbers in the results. Moreover, 60% of the rats with LPS died. This would indicate to the reviewer it is more of an infection during pregnancy model and the LPS dose is too high.

Prior to starting the animal study, we performed a comparison study to determine which immune-driven PE model best mimics human PE3 *in vivo* (PMID: 37842294). When examining and comparing rodent models of PE that are historically described in the literature, it is important that the existence of distinct PE disease subclasses is considered. As we, and others (PMID: 25679511, 22647886, 24477207, 12908998), have now clearly demonstrated the presence of distinct disease subclasses it is important to test and utilize rodent models that best mimic these subclasses. While there are several appropriate animal models for hypoxia-driven PE subclass (e.g. Reduced uterine perfusion pressure model), there was no specific model validated to study human PE3. Thus, in our previous study we assessed three commonly used models of chronic maternal inflammation (TNF- α infusion, Poly:IC treatment and LPS treatment) in relation to human inflammatory PE pathophysiology. In our laboratory settings, we found that only LPS treatment led to maternal BP increase compared to controls. We also assessed placental inflammation, mitochondrial impairment, and NAD⁺ dysregulation- the key features to identify an appropriate model for this disease subclass. We found that *in vivo*, inflammation-driven PE through LPS induction causes respiratory dysfunction of both mitochondrial complex I and II. LPS treatment also mimicked NAD⁺ related metabolic dysregulation in the placenta. Thus, we chose to use LPS-induced inflammation model. We used a dose of LPS previously reported to trigger maternal inflammation and induce high blood pressure (doi.org/10.1111/fcp.12501). We agree testing a lower dose of LPS would be worth examining in the future, to determine if it is still capable of inducing maternal hypertension in the absence of fetal loss. However, strikingly our data suggest that NR intervention can ameliorate such severe fetal impact of LPS treatment, demonstrating that NR is a compelling therapeutic option

to treat inflammation-driven PE. We have now added the mean blood pressure values in the text for GD 19. We agree the blood pressure for LPS model is not very high. But we would like to point out that in human PE3 disease, the maternal manifestation, including the degree of hypertension, is milder. Whereas the fetal manifestation of disease is exaggerated, with increased presence of small for gestational age and fetal growth restriction (PMID: 27160201, PMID: 30278173) – similar to the model used in the current study as well whereas fetal effect (FGR) is more frequently observed

2.6. Lines 411-414 - *"We hypothesized that there is a decline in placental NAD+ levels due to hyperactivity of NAD+ consuming enzymes and that it leads to impaired placental mitochondrial function, collectively contributing to the development of PE. Thus, replenishing NAD+ could be an attractive preventative strategy in this subclass of PE patients."* This should be in the rationale for the study not the methods

As suggested, we have now removed this from methods and added to introduction.

2.7. *"Placental mitochondrial dysfunction has long been recognized as a key component of PE pathophysiology [8-12]. Altered placental mitochondrial respiration and/or total content is observed in tissues collected from pregnancies that span the clinical spectrum of PE, associated with altered oxidative phosphorylation (OXPHOS) protein expression, mitochondrial structural damage, defects in fusion/fission dynamics and increased oxidative stress [8, 13-15]. There is strong evidence to suggest that placental mitochondrial dysfunction may be a common feature across all PE subclasses [8],"*

Placenta mitochondrial functionality assay and protein expression, and oxidative stress discussion should be in the methods not in the supplement.

We thank the reviewer for this suggestion. We have now moved the placenta mitochondrial functionality assay and protein expression, and oxidative stress section from the supplementary to methods.

2.8. I suggest authors discuss previous findings on TNF-alpha mediated mitochondrial damage in PE rat model (doi.org/10.1016/j.preghy.2021.02.006). It isn't in any of these references and it could support the studies with TNF.

We have now cited (doi.org/10.1016/j.preghy.2021.02.006) in the result section.

2.9. 10mg is really supraphysiological concentration of TNF alpha and is not what is seen in PE. The results state 10 ng/ml which is different than methods. Which was it?

We apologize for this discrepancy and any confusion this caused. In the in vitro work, 10 ng/ml of TNF-a was the concentration used. The inclusion of 10mg in the methods section was a typo. We thank the reviewer for pointing out this oversight. This has been corrected in the revised manuscript.

2.10. Some of the results repeat what is in the methods and could be reduced in the results section, especially with the rat model description.

The subtitles for the sections in results are way too long

Thank you for these editorial suggestions. We have now reduced the length of the text within the result section and shortened all subtitles.

2.11. An experiment demonstrating or a discussion on how reduced NAD⁺ levels would lead to reduced placental (or trophoblast) mitochondrial function is missing? Do authors believe SIRT6 are involved?

We have now added the following paragraph to the discussion.

“Currently we do not know the exact mechanism how NAD⁺ boosting improves placental mitochondrial function. Under proinflammatory conditions, we observed an improvement in ETC function with NR supplementation in trophoblast cells or placenta tissue but not in mitochondrial content. We speculate that since energy metabolism is dependent on NAD⁺ and its reduced form NADH, a decline in total NAD⁺ content will lead to a depressed energy metabolism, and NAD⁺ supplementation will prevent that. Moreover, activation of the SOD2/Sirt3 pathway could be a potential mechanism of reducing mitochondrial ROS and improving placenta function with NR supplementation, which needs further exploration.”

2.12. The link between PARP (DNA repair enzyme localized to nucleus) and mitochondrial function modulation via NAD could be better discussed.

In the introduction (lines 106-128), a paragraph has been dedicated to better describing the link between NAD⁺, PARP and mitochondrial dysfunction.

2.13. It is understandable authors chose TNF-alpha based on their previous microarray work, do you believe PARP activation is TNF-alpha mediated entirely or other inflammatory mediators recapitulate these findings?

We saw increase in other inflammatory cytokines in PE3 human placentas. Therefore, PARP activation is likely a result of the increase in several inflammatory cytokines. We think that other inflammatory mediators or their combination may also recapitulate these findings. In the current study we chose TNF-a as it has been used in several other studies to demonstrate inflammation-mediated trophoblast dysfunction.

2.14. A published study (doi.org/10.1091/mbc.E18-10-0650) shows PARP1 inhibits hexokinase (glycolysis), can you comment on your ECAR data in Fig 2G?

The aforementioned study suggests a decrease in glycolysis and increase dependency in OXPHOS with PARP1 activation when treated with methyl methanesulphonate (MMS) for 1-2hrs in Hela cells. We think it is important to consider that these observed results were based on very short-term treatments, and we cannot be certain what effects a longer treatment periods (i.e. 24hrs) would have yielded. In our in vitro study, which included an TNF-a treatment of 24hrs, we did not observe a significant decrease in ECAR. This could be due to several reasons, 1) TNF-a or inflammatory induction may function in different mechanisms than MMS mediated oxidative damage; 2) The observed findings could be timepoint dependent; or, 3) In different cell types the direct effect of PARP1 activation on metabolism may not be same.

2.15. Page 8, Line 169: Several studies demonstrate reduced COX-IV expression in preeclamptic patients. Hence, it may be not a suitable marker for mitochondrial content.

We have tested 2 typically used markers of mitochondrial content. One was citrate synthase, a mitochondrial matrix protein, and COX-IV, a mitochondrially encoded

electron transport chain complex. However, neither of them showed a significant difference in expression in the individual PE groups compared to controls.

2.16. Is there a reason you used different markers of DNA damage? 8-oxo-DG (human placenta) and p-H2A.X (rat placenta)? Measurement on mitochondrial oxidative damage would have been a good addition to establish mito damage in this study.

We had tested 8-oxoDG antibody in the rat placenta initially. However, we did not get a nuclear specific staining in the rat placenta. Therefore, we tested p-H2A.X antibody, which is an indirect indicator of DNA damage.

We agree that specifically determining mitochondrial oxidative damage would be more informative. However, it will require us to use freshly isolated mitochondria and perform ROS measurement assays. Currently we do not have the capacity to set up animal study to carry out such experiment.

2.17. Other studies in human PE placentas suggest that mt ROS is down, but there is no discussion of these findings compared the present work.

This is an interesting comment, as in our review of the literature we have discovered that most studies instead indicate *heightened* mtROS production (PMID: 32646252, PMID: 30315766, PMID: 30012871, PMID: 37237853) and *diminished* mitochondrial antioxidant activity (PMID: 30455461), with placental hypoxia contributing to this (PMID: 29778808). As there were no references associated with this reviewer comment we were unable to directly comment on the work highlighted in this comment.

2.18. Page 33, Fig 2: Did you validate increase in NAD⁺ levels after treating trophoblasts with NR?

Yes, this data is included in Supplementary figure 2.

2.19. Page 33, Fig 2H: Is there a reason complex I was not probed? Especially it being NADH dehydrogenase. Did you measure individual electron transport chain activities.

In the current study we used the OXPHOS antibody cocktail (Abcam, ab110413) to determine the expression of individual complexes. Using this antibody, clear bands for complex II, III, IV and V were apparent (Figure 2H, image shown above under comment 2.4). However, we were unable to clearly detect a band for complex I (supposed to appear below complex II in the figure above).

In vitro, we did not measure the activity of individual complexes. Instead, we performed the widely used seahorse technique to measure OXPHOS function on live cells.

2.20. Page 37: Did you measure electron transport chain complex expression and activities in rat placenta? (Can you please check it in supplemental?)

We have performed oxygraph respirometry assay to assess electron transport chain activity (Figure 4I-J). We have used OXPHOS antibody cocktail to determine the expression of individual complexes. We were able to determine the expression of complex II, III, and V (Supplementary Figure 4C). However, we could not detect complex I and IV with the antibody used. We have noted that others have described

challenges in accurately detecting or measuring the expression of these complexes in some tissue types considering its highly labile nature, being highly sensitive to the solubilization and heat incubation steps of protein preparation methods (PMID: 19806591, PMID: 17209039).

2.21. Is an impairment in mitochondria or the effects are just a reflection of metabolic shift.

Our data suggest that inflammation leads to decreased OXPHOS function in trophoblast and placenta. In our in vitro experiments, we did not observe any increase in glycolysis with TNF-alpha suggesting it may not be due to a metabolic shift.

2.22. Page 37, Fig 4E: Can you comment on why you didn't see increased NAD levels in NR vs. Saline group?

NAD⁺ can be quickly metabolized to its by product NAM. Therefore, we measured the levels of NAM by LC/MS confirming increased levels of this metabolite in placentas from NR-treated rats, as shown in the Supplementary figure 4A.

2.23. Can you comment on how PARP inhibitors would be an alternative approach to NAD⁺ replenishment?

We thank the reviewer for this suggestion. We have included a brief discussion of this in the revised manuscript.

September 4, 2024

RE: Life Science Alliance Manuscript #LSA-2023-02505-TR

Dr. Shannon A Bainbridge
University of Ottawa
Interdisciplinary School of Health Sciences
451 Smyth Rd, Rm 2116
Ottawa, Ontario K1H 8L1
Canada

Dear Dr. Bainbridge,

Thank you for submitting your revised manuscript entitled "NAD+ depletion is central to placental dysfunction in an inflammatory subclass of preeclampsia". We would be happy to publish your paper in Life Science Alliance pending final revisions necessary to meet our formatting guidelines.

- please be sure that the authorship listing and order is correct
- please upload all figure files individually, including the supplementary figure files; all figure legends should only appear in the main manuscript file. For publication, we require PowerPoint, TIFF, PDF, or EPS files.
- supplementary methods should be part of the main manuscript materials and methods section
- please add ORCID ID for the corresponding author -- you should have received instructions on how to do so
- please add a Summary Blurb/Alternate Abstract to our system
- please add the Twitter handle of your host institute/organization as well as your own or/and one of the authors in our system
- please upload Graphical Abstract separately with the file designation "Graphical Abstract."
- please add your main, supplementary figure, and table legends to the main manuscript text after the references section
- we encourage you to revise the figure legend for Figure S3 such that the figure panels are introduced in alphabetical order

FIGURE CHECKS:

- the scale bar is hard to see in Figure 1I
- please add a scale bar to Figure 2M
- is there a splice in Figure 2A, Stain free row, after the 5th column? If so, please indicate with a vertical black line and mention that the line indicates a splice in the figure legend.

A. FINAL FILES:

-- Summary blurb (enter in submission system): A short text summarizing in a single sentence the study (max. 200 characters including spaces). This text is used in conjunction with the titles of papers, hence should be informative and complementary to

the title. It should describe the context and significance of the findings for a general readership; it should be written in the present tense and refer to the work in the third person. Author names should not be mentioned.

B. MANUSCRIPT ORGANIZATION AND FORMATTING:

Sincerely,

Reviewer #2 (Comments to the Authors (Required)):

The authors either answered questions in my previous review or made the appropriate changes in the paper. I really have no further comments. The paper will add significant information to area of PE research

September 24, 2024

RE: Life Science Alliance Manuscript #LSA-2023-02505-TRR

Dr. Shannon A Bainbridge
University of Ottawa
Interdisciplinary School of Health Sciences
451 Smyth Rd, Rm 2116
Ottawa, Ontario K1H 8L1
Canada

Dear Dr. Bainbridge,

Thank you for submitting your Research Article entitled "NAD+ depletion is central to placental dysfunction in an inflammatory subclass of preeclampsia". It is a pleasure to let you know that your manuscript is now accepted for publication in Life Science Alliance. Congratulations on this interesting work.

DISTRIBUTION OF MATERIALS:

Again, congratulations on a very nice paper. I hope you found the review process to be constructive and are pleased with how the manuscript was handled editorially. We look forward to future exciting submissions from your lab.

Sincerely,
